# Geometry-Preserving Orthonormal Initialization for Low-Rank Adaptation in RLVR

**Ruijia Zhang** [1]   **Jiacheng Zhu** [2]   **Hanqing Zhu** [3]   **Laixi Shi** [4]

## Abstract

Low-rank adaptation (LoRA) and its variants enable parameter-efficient fine-tuning of large language models under the supervised fine-tuning (SFT) paradigm. However, their efficacy and behavior under Reinforcement learning with verifiable rewards (RLVR) are less well understood. In particular, two structurally initialized LoRA variants, PiSSA and MiLoRA, which outperform standard LoRA under SFT, can underperform standard LoRA under RLVR and may even exhibit training instability. These observations suggest that how to initialize the low-rank matrices in RLVR remains unclear. In this work, we develop a theoretical analysis of LoRA in RLVR, showing that orthonormal initialization achieves the minimal gap between LoRA's outcome and that of full fine-tuning. Guided by this insight, we propose geometry-preserving orthonormal initialization for low-rank adaptation in RLVR, leading to two new variants, LoRA-RLPO and LoRA-RLMO. Experiments on mathematical reasoning benchmarks show that the proposed orthonormal initialization stabilizes RLVR training and outperforms standard LoRA, contrasting with PiSSA and MiLoRA. Finally, our unified analysis for LoRA initialization also explains why PiSSA and MiLoRA can underperform in RLVR, which may be of independent interest. Code and checkpoints are publicly available at the repository.

## 1. Introduction

Large language models (LLMs) (Brown et al., 2020; Touvron et al., 2023) are typically pretrained on large-scale dataset via next-token prediction (Brown et al., 2020) and then fine-tuned on relatively smaller datasets to specialize for downstream applications. This paradigm has achieved remarkable success across diverse domains, including mathematical reasoning (Luo et al., 2025; Azerbayev et al., 2024), code generation (Rozière et al., 2024; Luo et al., 2024), healthcare (Singhal et al., 2023; Chen et al., 2023), and finance (Wu et al., 2023; Yang et al., 2025). Because fine-tuning is far more accessible than pretraining a new LLM, it has attracted substantial interest in the community. While fine-tuning all parameters in an LLM ("full fine-tuning") is natural, it is practically highly memory-intensive: fully fine-tuning even a 7B model can require over 100GB of GPU memory (Dettmers et al., 2023). This high resource demand limits accessibility for practitioners and motivates parameter-efficient fine-tuning (PEFT) methods, which update only a small subset of parameters while keeping the base model frozen (Houlsby et al., 2019; Li and Liang, 2021). Among them, Low-Rank Adaptation (LoRA) (Hu et al., 2022) is widely used due to its efficiency and ease of implementation. For any weight matrix $W_0 \in \mathbb{R}^{m \times n}$ in a pretrained model, LoRA parameterizes the update as $\Delta W^{\text{lora}} = BA$ with $B \in \mathbb{R}^{m \times r}$ and $A \in \mathbb{R}^{r \times n}$, where $r$ is much smaller than $\min(m, n)$. This significantly reduces the number of parameters under training, while still yielding a dense matrix $W_0 + \Delta W^{\text{lora}}$ at inference time.

Beyond supervised fine-tuning (SFT), a widely used LLM fine-tuning paradigm that trains on high-quality question–response pairs, reinforcement learning with verifiable rewards (RLVR) has recently emerged as a pivotal paradigm, proving effective across tasks such as mathematical reasoning and coding (Guo et al., 2025; Shao et al., 2024). RLVR uses rule-based feedback (e.g., answer correctness) instead of learned reward models. However, RLVR incurs substantially higher memory costs than SFT, as it requires keeping a reference model in memory to compute KL divergence (Ziegler et al., 2020; Zhou et al., 2024) and storing multiple responses per prompt for group-based advantage estimation (Shao et al., 2024). This makes LoRA and its

---
[1]Department of Applied Mathematics and Statistics, Johns Hopkins University, MD, USA [2]Meta Superintelligence Labs [3]Department of Electrical and Computer Engineering, The University of Texas at Austin, TX, USA [4]Department of Electrical and Computer Engineering, Johns Hopkins University, MD, USA. Correspondence to: Laixi Shi <laixis@jhu.edu>.

*Proceedings of the 43$^{rd}$ International Conference on Machine Learning*, Seoul, South Korea. PMLR 306, 2026. Copyright 2026 by the author(s).

memory-efficient variants particularly attractive for RLVR, especially given that LoRA has already shown strong potential in this setting, matching full fine-tuning in certain cases (Schulman and Lab, 2025).

Despite this progress, the behavior of LoRA and its structural variants under RLVR remains less understood than under SFT, limiting further advances in low-rank fine-tuning for RL. In particular, how to initialize the low-rank matrices $B$ and $A$ is increasingly unclear in light of recent observations. PiSSA (Meng et al., 2024) and MiLoRA (Wang et al., 2025), two LoRA variants that improve performance and accelerate convergence in SFT, can underperform standard LoRA under RLVR and may even exhibit training instability (Yin et al., 2025). Both methods initialize $B$ and $A$ via the singular value decomposition (SVD) of pretrained weights, but in opposite directions: PiSSA uses the top-$r$ principal singular directions, while MiLoRA targets the bottom-$r$ tail directions. In addition, prior work suggests that, due to the KL constraint, RLVR updates are encouraged to stay close to the reference policy (Wu et al., 2026; Shenfeld et al., 2025) and may favor off-policy subspaces that differ from those preferred by SFT (Zhu et al., 2025). This discrepancy between RLVR and SFT likely reflects their distinct optimization dynamics. Consequently, LoRA design principles developed for SFT are no longer guaranteed to transfer to RLVR, leaving the appropriate initialization and subspace choice in RLVR an open question. In this work, we focus on:

*What initialization is effective for low-rank adaptation in RLVR fine-tuning?*

To this end, we provide a rigorous analysis demonstrating that by initializing $B = 0$ in accordance with standard LoRA, orthonormal initialization for $A$ is potentially optimal and yields superior performance in practice. Our primary contributions are as follows:

- **Orthonormal initialization towards optimal.** To understand the behavior of LoRA, we provide a theoretical analysis of LoRA's optimization dynamics, showing that orthonormal initialization of $A$ with $B = 0$ minimizes the gap between LoRA and full fine-tuning (Theorem 4.2).

- **Geometry-preserving orthonormal initialization.** Motivated by this result, we propose LoRA-RLPO and LoRA-RLMO, SVD-based initializations for $A$ that remain orthonormal while preserving geometric information from pretrained weights. Both methods outperform standard LoRA in RLVR in the conducted experiments.

- **Insights into LoRA variants' failures in RLVR.** Our analysis framework offers a unified explanation for the instability of SVD-based methods such as PiSSA and

MiLoRA in RLVR. Their failures stem from two coupled factors: **subspace geometry**, which accelerates updates along specific directions, and **singular value scaling**, which amplifies update magnitudes. Together, these induce aggressive optimization trajectories that rapidly violate the implicit KL constraint, destabilizing training regardless of whether principal or minor singular directions are targeted.

**Conflict of Interest Disclosure.** Jiacheng Zhu is employed by Meta Superintelligence Labs. This work does not evaluate a proprietary model developed by Meta. The authors declare no other financial conflicts of interest.

## 2. Background

In this section, we formalize the fine-tuning problem for LLMs. Consider an LLM parameterized by $\theta = \{W^{(\ell)}\}_{\ell=1}^{L}$, the collection of weight matrices across its $L$ layers (e.g., fully connected and attention layers). Without loss of generality and a slight abuse of notation, we focus on a single weight matrix $W$ in the following discussion. Full fine-tuning optimizes the model parameters $\theta$ by updating all weight matrices. Specifically, for any pretrained weight matrix $W_0 \in \mathbb{R}^{m \times n}$, full fine-tuning learns $W = W_0 + \Delta W^{\text{full}}$, where the update $\Delta W^{\text{full}} \in \mathbb{R}^{m \times n}$ is unconstrained, so as to minimize a task-specific loss $\mathcal{L}(\theta) = \mathcal{L}(\{W^{(\ell)}\}_{\ell=1}^{L})$. This approach requires storing full gradients and optimizer states for all weight matrices, which can become prohibitive for large-scale models.

### 2.1. Low-rank adaptation (LoRA) and variants

We first review the LoRA algorithm (Hu et al., 2022). Consider a pretrained weight matrix $W_0 \in \mathbb{R}^{m \times n}$, LoRA parameterizes the weight update as

$$W = W_0 + \Delta W^{\text{lora}}, \qquad \text{with} \quad \Delta W^{\text{lora}} = BA, \quad (1)$$

where $B \in \mathbb{R}^{m \times r}$ and $A \in \mathbb{R}^{r \times n}$ are low-rank matrices with rank $r$ much smaller than $m$ and $n$ ($r \ll \min\{m, n\}$). Consequently, the optimization of $\Delta W^{\text{lora}}$ is restricted to a low-rank subspace of $\mathbb{R}^{m \times n}$. The initialization is set as follows:

$$B_0 = 0_{m \times r}, \quad A_0 \sim \mathcal{N}\left(0, \tfrac{1}{n}\right)^{r \times n}. \quad (2)$$

**SVD-based initialization variants.** Beyond the initialization in (2), many prior works propose alternative initializations for low-rank fine-tuning. We describe two representative SVD-based variants below. Let $W_0 = U\Sigma V^{\top}$ be the singular value decomposition, where $U \in \mathbb{R}^{m \times k}$, $\Sigma = \text{diag}(\sigma_1, \ldots, \sigma_k)$ with $\sigma_1 \geq \sigma_2 \geq \cdots \geq \sigma_k > 0$, and $V \in \mathbb{R}^{n \times k}$. Here, $k = \text{rank}(W_0)$, i.e., the number of positive singular values of $W_0$.

## Initialization

*Figure 1.* Comparison of LoRA initialization strategies. LoRA uses random Gaussian $A_0$ with $B_0 = 0$. PiSSA and MiLoRA initialize both adapters from the principal and minor singular components of $W_0$, respectively, with $B_0 \neq 0$ and singular value scaling. Our proposed methods, LoRA-RLPO and LoRA-RLMO, initialize orthonormal $A_0$ from the principal and minor right singular vectors with $B_0 = 0$.

**PiSSA** (Meng et al., 2024) initializes with top-$r$ principal components as follows:

$$B_0 = U_r \Sigma_r^{1/2}, \quad A_0 = \Sigma_r^{1/2} V_r^\top \quad (3)$$

where $U_r$ and $V_r$ denote the first $r$ columns of $U$ and $V$, respectively, and $\Sigma_r$ is the $r \times r$ diagonal matrix of the top $r$ singular values.

**MiLoRA** (Wang et al., 2025) initializes using the bottom-$r$ minor components:

$$B_0 = U_{-r} \Sigma_{-r}^{1/2}, \quad A_0 = \Sigma_{-r}^{1/2} V_{-r}^\top, \quad (4)$$

where $U_{-r}$ and $V_{-r}$ denote the last $r$ columns of $U$ and $V$, respectively, and $\Sigma_{-r}$ is the $r \times r$ diagonal matrix of the bottom $r$ singular values.

**OLoRA** (Büyükakyüz, 2024) initializes with top-$r$ principal components as follows:

$$B_0 = U_r, \quad A_0 = V_r^\top. \quad (5)$$

All three methods then replace each pretrained matrix $W_0$ with the residual $W_0 - B_0 A_0$, which is kept frozen, and optimize only $BA$ thereafter. Equivalently, the effective weight is parameterized as $W = (W_0 - B_0 A_0) + BA$, ensuring $W = W_0$ at initialization regardless of which singular components are used.

### 2.2. Finetuning paradigms: SFT vs. RLVR

Besides the parameter-update setting, we now introduce two widely used fine-tuning frameworks and their corresponding objective functions.

**Supervised fine-tuning (SFT).** Consider a supervised dataset $\mathcal{D} = \{(q_i, a_i^\star)\}_{i=1}^N$ consisting of many input-output pairs, where $q_i$ denotes a prompt or instruction and $a_i^\star$ is the corresponding ground-truth response. The SFT process minimizes cross-entropy against ground-truth labels as follows:

$$\mathcal{L}_{\text{SFT}}(\theta) := -\mathbb{E}_{(q,a^*) \sim \mathcal{D}} \left[ \log \pi_\theta(a^*|q) \right].$$

SFT imposes no explicit constraint on the weight movement, allowing the parameters to drift arbitrarily far from $W_0$.

**Reinforcement learning with verifiable rewards (RLVR).** RLVR fine-tunes large language models with RL using automatically verifiable, rule-based rewards $R$ (e.g., exact-match correctness on math or code), thereby eliminating the need for a learned reward model. In this work, we focus on DAPO (Yu et al., 2026), a representative RLVR algorithm for long-CoT reasoning. We choose DAPO because it provides a fully open-source large-scale RL system with publicly available algorithmic details, code infrastructure, and training recipe, making it a reproducible and widely adopted testbed for studying RLVR optimization (Yu et al., 2026). Moreover, recent studies and implementations of parameter-efficient RLVR commonly evaluate LoRA-style adaptation under DAPO or DAPO-like clipped policy optimization, making it a natural setting for analyzing how low-rank initialization affects RLVR stability (Yin et al., 2025).

Specifically, DAPO samples a group of outputs $\{o_i\}_{i=1}^G$ for each question $q$ paired with answer $a$, and updates the policy by optimizing the following clipped importance-ratio

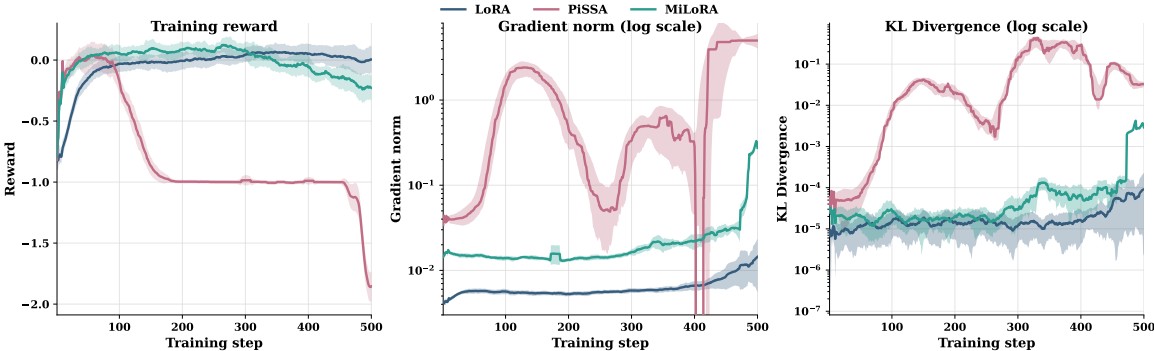

*Figure 2.* Training dynamics of RLVR via DAPO on benchmark DAPO-MATH. Left: Training reward comparison. Middle: The aggregate Frobenius norm of gradients over trainable parameters. Right: KL divergence during training. Both PiSSA and MiLoRA exhibit training reward collapse, higher gradient norm and KL divergence than standard LoRA.

objective:

$$\mathcal{L}^{\text{DAPO}}(\theta^+) = \mathbb{E}_{(q,a)\sim\mathcal{D}, \{o_i\}\sim\pi_\theta(\cdot|q)} \left[ \frac{1}{\sum_{i=1}^{G}|o_i|} \sum_{i=1}^{G} \sum_{t=1}^{|o_i|} \right.$$

$$\left. \min\left( r_{i,t}(\theta^+)\hat{A}_{i,t}, \text{ clip}\left(r_{i,t}(\theta^+), 1-\epsilon_{\text{low}}, 1+\epsilon_{\text{high}}\right)\hat{A}_{i,t}\right)\right],$$

$$\text{s.t.} \quad 0 < \left|\{o_i \mid \text{is\_equivalent}(a, o_i)\}\right| < G,$$

$$(6)$$

where $r_{i,t}(\theta^+) = \frac{\pi_{\theta^+}(o_{i,t}|q,o_{i,<t})}{\pi_\theta(o_{i,t}|q,o_{i,<t})}$, $\hat{A}_{i,t} = \frac{R_i - \text{mean}(\{R_i\}_{i=1}^G)}{\text{std}(\{R_i\}_{i=1}^G)}$. A key ingredient of DAPO is the clipped importance ratio, which constrains $r_{i,t}(\theta^+)$ to $[1-\epsilon_{\text{low}}, 1+\epsilon_{\text{high}}]$ and thereby implicitly limits the policy drift between $\pi_{\theta^+}$ and $\pi_\theta$.

**Training stability demands constrained KL divergence.**
The conservative-update principle implemented by DAPO's clipped importance ratio is not unique to DAPO: it underlies many popular policy-gradient algorithms (e.g., TRPO Schulman et al. (2015) and PPO Schulman et al. (2017)) as well as RLVR popular variants such as GRPO (Shao et al., 2024). By preventing the updated policy from deviating too far from the current one in a single step, the clipping mechanism can be viewed as a surrogate method to constrain the following KL divergence between $\pi_{\theta^+}$ and $\pi_\theta$ (Kakade and Langford, 2002) in a safe region:

$$D_{\text{KL}}(\pi_{\theta^+} \| \pi_\theta) =$$
$$\mathbb{E}_{q\sim\mathcal{D}, y\sim\pi_{\theta^+}(\cdot|q)} \left[ \sum_t \log \frac{\pi_{\theta^+}(y_t \mid q, o_{<t})}{\pi_\theta(y_t \mid q, o_{<t})} \right]. \quad (7)$$

Consequently, an excessively large $D_{\text{KL}}(\pi_{\theta^+} \| \pi_\theta)$ may violate this implicit trust-region constraint and can cause performance degradation or training collapse, since the surrogate objective is no longer guaranteed to be a lower bound of the true reward objective (Kakade and Langford, 2002). We therefore adopt $D_{\text{KL}}(\pi_{\theta^+} \| \pi_\theta)$ as a key diagnostic for the

training stability of RLVR, and report its sample estimate over response tokens in the empirical analyses that follow.

## 3. Instability of SVD-Based LoRA Initializations in RLVR

Noting that the behavior of LoRA variants in RLVR remains underexplored, limiting further advances in low-rank fine-tuning for many tasks such as reasoning. In this work, we focus on studying the *initialization* module of LoRA for RLVR, focusing on two prominent variants (PiSSA and MiLoRA) that have demonstrated strong performance in supervised fine-tuning (SFT). While both PiSSA and MiLoRA outperform standard LoRA in SFT, they instead underperform standard LoRA under RLVR and even exhibit clear training collapse, as shown in Figure 2 (left). To understand the source of this failure, we monitor the introduced KL divergence in (7) and the Frobenius norm of the gradient for training stability, as shown in Figure 2 (middle and right). Both PiSSA and MiLoRA incur substantially larger gradient norms and higher cumulative KL divergence than standard LoRA throughout training, indicating markedly unstable optimization processes. These observations raise a natural question: *Why do LoRA initialization principles that succeed in SFT, such as those of PiSSA and MiLoRA, break down under RLVR?*

**Initialization of LoRA largely governs final optimization outcome.** To answer the question, we begin by analyzing how the initialization step shapes the subsequent optimization trajectory and the resulting fine-tuned model. To this end, we visualize the post-training update distribution and cumulative energy after RLVR fine-tuning in Figure 3. Recall that PiSSA (cf. (3)) and MiLoRA (cf. (4)) initialize LoRA in the top and bottom singular-vector subspaces, respectively, with singular value scaling through $\Sigma_r^{1/2}$ or $\Sigma_{-r}^{1/2}$. As shown in Figure 3, the update magnitudes and final energy distributions of the fine-tuned outputs closely

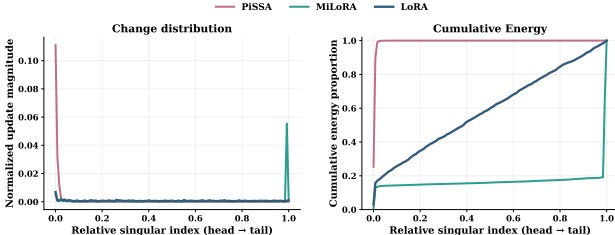

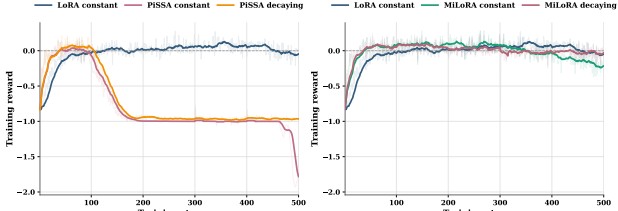

*Figure 3.* **SVD-aligned update distribution and cumulative energy after RLVR training.** For each method, we analyze the trained LoRA update $\Delta W = \frac{\alpha}{r} BA$ on the query projection and attention output projection matrices from Transformer layers 0, 14, and 27 of the 28-layer model, where each projection weight has size $W \in \mathbb{R}^{1536 \times 1536}$. Given the singular value decomposition of the frozen pretrained weight $W = U\Sigma V^\top$, we measure the update magnitude along the pretrained singular directions as $c_i = |u_i^\top \Delta W v_i|$, and normalize it by $p_i = c_i / \sum_j c_j$. The left panel plots $p_i$ over the relative singular-mode index, ordered from the largest to smallest singular values of $W$, while the right panel plots the corresponding cumulative energy $\sum_{j \le i} c_j^2 / \sum_j c_j^2$. Curves are averaged over the six analyzed projection matrices. The distinct patterns indicate that initialization affects not only the starting point of optimization, but also the spectral structure of the updates learned during RLVR.

mirror these initialized subspaces: PiSSA produces substantially larger updates along the top singular-vector directions, MiLoRA concentrates its updates along the bottom singular-vector directions, and in both cases the fine-tuned outcomes retain high cumulative energy in the corresponding spectral regions. This demonstrates that LoRA initialization does *not* merely set the optimization starting point, but rather it intrinsically governs the entire optimization trajectory and, consequently, the final outcome.

### 3.1. Instability sources of LoRA family in RLVR.

Another key observation from Figure 3 is that, after geometry-informed initialization, the final learned updates remain highly concentrated along the predefined singular directions, producing aggressively amplified weight changes. Such directionally intensified updates likely drive the large gradient norms and rapid KL growth observed under RLVR in Figure 2, which in turn contribute to the performance collapse of PiSSA and MiLoRA. To pinpoint the source of this instability, we disentangle three potential factors: (1) the learning rate during optimization, and two arising from initialization, namely, (2) the selected singular subspace, which determines the update direction, and (3) the singular value scaling, which amplifies updates along that direction.

**Learning rate partially controls update magnitude.** To assess the role of the learning rate in driving instability, we conduct an ablation comparing constant and decaying schedules, shown in Figure 4. Slowing optimization through learning rate decay offers partial mitigation for both PiSSA

*Figure 4.* **Ablation on learning-rate decay for PiSSA and MiLoRA under RLVR training.** The left panel compares PiSSA with a constant learning rate and a cosine-decaying schedule, together with the LoRA constant-learning-rate baseline; the right panel shows the analogous comparison for MiLoRA. Both methods benefit from slower optimization, confirming that enlarged effective updates are a primary source of instability in RLVR training. Nevertheless, while MiLoRA with learning-rate decay approaches the stable behavior of standard LoRA, PiSSA still exhibits significant late-stage collapse, pointing to an inherently more hazardous optimization direction.

and MiLoRA: as Figure 4 reveals, MiLoRA with a decaying learning rate approaches the stable behavior of standard LoRA. PiSSA, however, still suffers severe collapse in the later phases of training, indicating that learning rate decay alone does not resolve the underlying instability. This points to an inherently vulnerable optimization landscape for PiSSA: once updates are initialized and accelerated along its geometry-informed directions, the optimization trajectory tends to exit the stable optimization regime.

**Singular value scaling destabilizes training across singular subspaces.** Beyond the choice of subspace, singular value scaling is another potentially critical factor for optimization stability. To isolate its effect from subspace selection, we use OLoRA as a controlled baseline: since OLoRA and PiSSA target the same principal subspace of $W_0$, any difference in stability can be attributed to the *singular value scaling* inherent to PiSSA. We investigate this effect from both theoretical and empirical perspectives.

*Theoretical analysis.* We first present a theorem that quantifies how PiSSA's initialization induces excessive update magnitudes compared to OLoRA.

**Theorem 3.1** (PiSSA Gradient Amplification). *The first-step weight updates $\Delta W_1^{PiSSA}$ of PiSSA and $\Delta W_1^{OLoRA}$ of OLoRA satisfy:*

$$\frac{\|\Delta W_1^{PiSSA}\|_F}{\|\Delta W_1^{OLoRA}\|_F} \ge \sigma_r,$$

*where $\sigma_r$ is the $r$-th largest singular value of $W_0$.*

The proof is deferred to Appendix D. For pretrained LLMs, singular values typically follow a heavy-tailed distribution with $\sigma_r \gg 1$ at moderate rank $r$. Theorem 3.1 therefore implies that PiSSA scales up weight updates by a substantial factor relative to OLoRA. To proceed, we invoke a result

from Zhu et al. (2025), which shows that enforcing a KL constraint $D_{\mathrm{KL}}(\pi_{\theta+} \| \pi_\theta)$ keeps the updated policy close to the current policy and, in turn, bounds the magnitude of the corresponding weight change.

**Theorem 3.2** (KL constraint implies weight bound (Zhu et al., 2025, Gate I)). *Assume* $\log \pi_\theta$ *is* $C^3$*, where* $F(\theta)$ *denotes the Fisher information matrix* [1]*. Consider a single-step update of the model parameters from* $\theta$ *to* $\theta^+$*, so that each weight matrix* $W \subset \theta$ *is updated to* $W + \Delta W$*. Suppose that the KL divergence term in* (7) $D_{\mathrm{KL}}(\pi_{\theta+} \| \pi_\theta) \leq K$ *and that, on the update subspace,* $F(\theta) \succeq \mu I$ *for some* $\mu > 0$*. Then, for* $K$ *sufficiently small, every weight block* $W \subset \theta$ *satisfies* $\|\Delta W\|_F \leq \sqrt{\frac{2K}{\mu}} \left(1 + o(1)\right)$*.*

Theorem 3.2 shows that when the KL divergence is constrained by $K$, a corresponding bound is imposed on every weight update, $\|\Delta W\|_F \lesssim \sqrt{2K/\mu}$. Consequently, an excessively large $\|\Delta W\|_F$ will drive $D_{\mathrm{KL}}(\pi_{\theta+} \| \pi_\theta)$ beyond the budget $K$, potentially violating the conservative-update requirement and destabilizing training (see Section 2.2). As Theorem 3.1 establishes, PiSSA's singular-value scaling amplifies the weight update $\|\Delta W_1^{\mathrm{PiSSA}}\|_F$ along the principal spectral directions relative to OLoRA's $\|\Delta W_1^{\mathrm{OLoRA}}\|_F$, making it more prone to such violation and leaving the safe KL region and therefore more vulnerable to training instability in RLVR.

*Empirical analysis.* Figure 5 confirms this analysis in the principal singular subspace. In the first column, the top panel reports training reward and the bottom panel reports KL divergence between the rollout policy and the current policy for LoRA, PiSSA, and OLoRA. Compared with PiSSA, OLoRA incurs substantially smaller KL divergence, which by Theorem 3.2 implies a correspondingly smaller weight update magnitude. This reduced policy drift partially mitigates the reward collapse observed in PiSSA.

To verify that this destabilizing effect is not unique to the principal subspace, we extend the ablation to the minor singular directions via OLoRA-tail. As a counterpart to MiLoRA, OLoRA-tail targets the same tail singular subspace but removes singular value scaling:

$$B_0 = U_{-r}, \quad A_0 = V_{-r}^\top,$$

where $U_{-r}$ and $V_{-r}$ denote the last $r$ columns of $U$ and $V$, respectively. As shown in the second column of Figure 5, removing singular value scaling in the tail subspace yields markedly more stable dynamics and closely mirrors standard LoRA.

Taken together, the comparisons in Figure 5 reveal a nuanced interaction between subspace geometry and singular

---

[1]Here $C^3$ means having continuous derivatives up to order 3. The Fisher information matrix is defined as $F(\theta) = \mathbb{E}_{x \sim \mathcal{D}, y \sim \pi_\theta(\cdot|x)}[\nabla_\theta \log \pi_\theta(y|x) \nabla_\theta \log \pi_\theta(y|x)^\top]$.

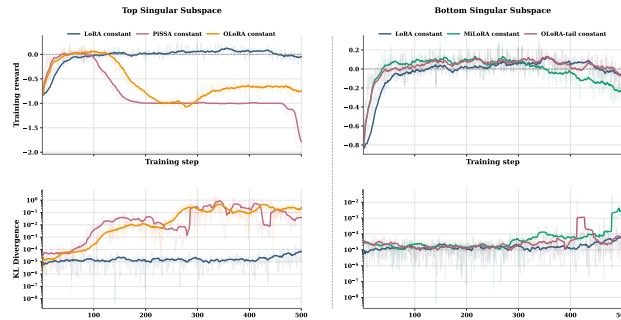

*Figure 5.* **Singular value scaling exacerbates instability beyond subspace selection.** We compare constant-learning-rate DAPO training dynamics for LoRA, PiSSA, OLoRA, MiLoRA, and OLoRA-tail. The first column compares methods associated with the top singular subspace (LoRA, PiSSA, and OLoRA), while the second column compares methods associated with the bottom singular subspace (LoRA, MiLoRA, and OLoRA-tail). The first row reports training reward, and the second row reports KL divergence between the rollout policy and the current policy. Within each subspace, removing singular value scaling, from PiSSA to OLoRA for the top subspace and from MiLoRA to OLoRA-tail for the bottom subspace, mitigates reward collapse and reduces policy drift. These controlled comparisons show that singular value scaling is a distinct source of RLVR instability, beyond the choice of singular subspace.

value scaling. In the principal singular directions, OLoRA delays and weakens the reward collapse observed in PiSSA. In the minor singular directions, OLoRA-tail nearly eliminates the instability observed in MiLoRA, yielding reward and KL dynamics close to standard LoRA. Thus, although certain singular subspaces are more prone to exceeding the KL leash, removing singular value scaling and enforcing orthonormality provides a viable path to more stable RLVR training. This motivates a deeper study of LoRA optimization dynamics and a principled initialization strategy that utilizes the geometry of singular subspace while avoiding destabilizing singular value scaling.

## 4. Geometry-Preserving Orthonormal Initialization for LoRA

In this section, we formally analyze the optimization dynamics of LoRA to investigate initialization strategy, studying how initialization can bridge the gap to full fine-tuning while maintaining training stability. The proof has been deferred to the Appendix C. Based on this analysis, we propose orthonormal initialization strategy that preserves the geometry of the pretrained weight space while avoiding singular value scaling.

### 4.1. Optimization Dynamics of LoRA

Since LoRA constrains updates to a rank-$r$ subspace with $r < n$, it often cannot exactly match full fine-tuning. We

quantify this approximation gap with respect to the initialization strategy in Theorem 4.2, aiming to reduce the gap as much as possible.

**Orthonormal initialization in LoRA leads closer to full fine-tuning.** Without loss of generality, we consider a general RLVR objective:

$$\mathcal{L}_{\text{RLVR}}(\theta) := \mathbb{E}_{x \sim \mathcal{D}, \, y \sim \pi_\theta(\cdot|x)} [R(x,y)] - \beta \cdot \text{KL}(\pi_\theta \| \pi_{\text{ref}}), \tag{8}$$

where the reward $R(x,y) \in \{0,1\}$ equals 1 if $y$ is a valid solution to $x$ and 0 otherwise, $\pi_{\text{ref}}$ is some reference policy, and $\beta > 0$ denotes a KL penalty coefficient Following standard LoRA, we let $B_0 = 0$ and consider a more general initialization for $A_0 \in \mathbb{R}^{r \times n}$. Let $T$ denote the total number of training iterations and $t \in \{0, 1, \ldots, T\}$ the current iteration. Consider a input $x$ and a single linear layer with weight matrix $W_t = W_0 + B_t A_t$. The forward pass computes logits $z_t = W_t x$, and the policy $\pi_\theta(y \mid x) = \text{softmax}(z_t)$ gives the probability of generating output $y$, where $\theta$ denotes the model parameters including $W_t$. At time $t$, the gradients with respect to the two LoRA matrices are $\frac{\partial \mathcal{L}}{\partial A} = B_t^\top G_t, \frac{\partial \mathcal{L}}{\partial B} = G_t A_t^\top$, where $G_t = \nabla_{W_t} \mathcal{L}$ denotes the gradient of the loss with respect to $W_t$.

**Assumption 4.1.** The RLVR loss $\mathcal{L}$ from (8) is $L$-smooth with respect to logits $z = Wx$. The gradient is bounded: $\|G_t\|_F \leq M$ for all $t$.

The assumption on the loss holds for the RLVR objective with binary rewards, with a detailed derivation provided in Appendix B.2. The assumption on the gradient is typically enforced via the clipped importance-ratio mechanism, which is widely used in DAPO, GRPO, and other RL algorithms.

**Theorem 4.2** (LoRA Approximation Error). *Let $W_T^{\text{full}}$ and $W_T^{\text{LoRA}}$ be the weights after $T$ steps of full fine-tuning and LoRA under RLVR. Under Assumptions 4.1, one has*

$$\frac{1}{T} \|W_T^{\text{LoRA}} - W_T^{\text{full}}\|_F \leq \frac{M\eta}{1 - L\eta\|x\|_2^2} \|I_n - A_0^\top A_0\|_2 + \mathcal{O}(\eta^2).$$

*Furthermore, for $A_0 \in \mathbb{R}^{r \times n}$ with $r < n$, $\|I_n - A_0^\top A_0\|_2 \geq 1$ holds with equality if $A_0$ has orthonormal rows, i.e., $A_0 A_0^\top = I_r$.*

Theorem 4.2 shows that when the learning rate $\eta$ is sufficiently small, typically satisfied in practice, the approximation error between LoRA and full fine-tuning is controlled by $\|I_n - A_0^\top A_0\|_2$. Furthermore, orthonormal initialization of $A_0$ minimizes this term to 1. With standard LoRA initialization $B_0 = 0$, Theorem 4.2 implies that orthonormal initialization $A_0$ minimizes the approximation gap between LoRA and full fine-tuning in RLVR, making LoRA's performance closer to that of full fine-tuning.

**Orthonormal initialization stabilizes RLVR training.** Beyond minimizing the gap between LoRA and full fine-tuning, orthonormal initialization also controls the update magnitude at each step, which is critical for stable training under RLVR.

**Proposition 4.3** (Bounded Weight Updates for Orthonormal Initialization). *For LoRA parameterization $\Delta W^{lora} = BA$ with $B_0 = 0$ and orthonormal $A_0$, the first-step LoRA update satisfies*

$$\|\Delta W_1^{lora}\|_F = \eta \|G_0 A_0^\top A_0\|_F \leq \eta \|G_0\|_F,$$

*where $G_0 = \nabla_{W_0} \mathcal{L}$.*

This result shows that, at the first iteration, orthonormal initialization guarantees that the LoRA weight update $\|\Delta W_1^{\text{lora}}\|_F$ does not exceed that of full fine-tuning in Frobenius norm. This controlled first-step behavior is likely to persist throughout training by extending the proof of Proposition 4.3, thereby potentially mitigating abrupt policy shifts and improving RLVR fine-tuning stability.

## 4.2. Geometry-Preserving Orthonormal Initialization for LoRA

The theoretical insights above indicate that orthonormal initialization of $A_0$ is beneficial for low-rank fine-tuning in RLVR, both in minimizing the gap to full fine-tuning and in ensuring bounded weight updates. Motivated by this, we propose two initialization schemes that enforce orthonormality of $A_0$ while setting $B_0 = \mathbf{0}_{m \times r}$, illustrated in Figure 1 alongside comparisons to prior works. To preserve the geometric structure of the pretrained weight matrix $W_0$, both schemes are derived from its SVD.

- **Principal orthonormal initialization** (LoRA-RLPO). We initialize the adapter $A_0$ using the principal singular vectors:
$$B_0 = \mathbf{0}_{m \times r}, \quad A_0 = V_r^\top,$$
where $V_r \in \mathbb{R}^{n \times r}$ contains the top-$r$ right singular vectors of $W_0$. This preserves the geometric information of the pretrained model by aligning the adapter with the principal directions of $W_0$. The design is similar in spirit to PiSSA, as both retain the top-$r$ singular directions of $W_0$; however, PiSSA additionally incorporates the singular value scaling.

- **Minor orthonormal initialization** (LoRA-RLMO). Analogously, we define an initialization targeting the minor subspace:
$$B_0 = \mathbf{0}_{m \times r}, \quad A_0 = V_{-r}^\top,$$
where $V_{-r} \in \mathbb{R}^{n \times r}$ consists of the bottom-$r$ right singular vectors of $W_0$. While MiLoRA also targets the

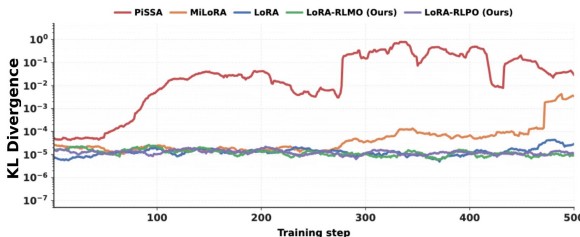

*Figure 6.* KL divergence during training for different initialization methods. PiSSA shows the highest KL divergence, MiLoRA exhibits intermediate KL growth, and LoRA remains relatively stable. Our proposed methods (LoRA-RLMO and LoRA-RLPO) maintain the lowest KL trajectories overall, indicating improved training stability.

minor subspace, it incorporates singular-value scaling and a nonzero $B_0$; in contrast, LoRA-RLMO adopts an orthonormal $A_0$ with $B_0 = \mathbf{0}$.

## 5. Experiments and Analysis

In this section, we conduct experiments to validate our theoretical findings across various benchmarks.

**Experimental setup.** We fine-tune DeepSeek-R1-Distill-Qwen-1.5B using DAPO (Yu et al., 2026) on DAPO-Math-17k (Yu et al., 2026) with rank $r = 16$ LoRA applied to all linear layers. We compare standard LoRA, PiSSA, MiLoRA, LoRA-RLPO, and LoRA-RLMO across five mathematical reasoning benchmarks: GSM8K (Cobbe et al., 2021) (1,319 samples), MATH500 (Hendrycks et al., 2021) (500 samples, mean@4), and AIME 2022/2023/2024 (30 samples each, mean@32) (Zhang and Math-AI, 2024). Full experimental details are provided in Appendix E.

**Performance.** Table 1 compares SVD-based LoRA initialization methods at step 500 across five mathematical reasoning benchmarks, reporting the mean and standard deviation over multiple seeds. Overall, LoRA-RLPO achieves the highest average accuracy ($65.03_{\pm 0.55}\%$), followed by LoRA-RLMO ($63.76_{\pm 1.64}\%$) and LoRA ($62.40_{\pm 2.96}\%$). LoRA-RLPO obtains the best performance on MATH500, AIME22 , AIME23, and AIME24 , while LoRA-RLMO achieves the strongest GSM8K result. MiLoRA consistently trails behind LoRA on average, and PiSSA performs substantially worse than the other methods in this setting. These results show that our geometry-preserving orthonormal initializations improve the stability and effectiveness of SVD-informed LoRA variants for RLVR.

**Training stability.** Figure 6 monitors and compares the KL divergence term $D_{\mathrm{KL}}(\pi_{\theta+} \| \pi_\theta)$ in (7) during training for different initialization methods. PiSSA shows the largest KL divergence by a wide margin, while MiLoRA exhibits

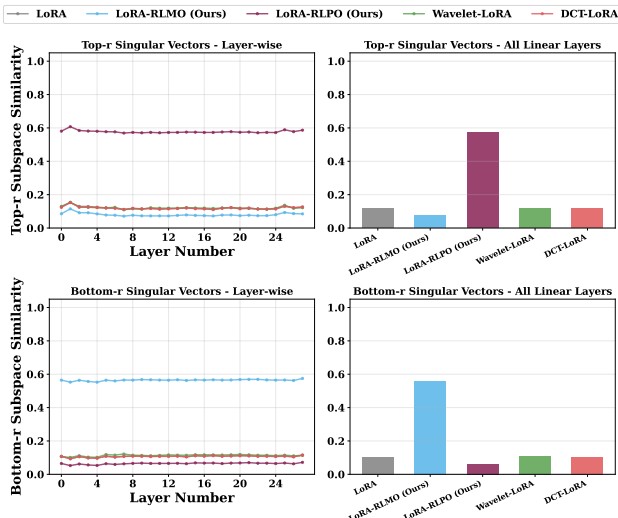

*Figure 7.* Subspace similarity between learned adapters and singular vectors of pretrained weights $W_0$. Top row: similarity with principal (top-$r$) right singular vectors. Bottom row: similarity with minor (bottom-$r$) right singular vectors. Left column shows per-layer similarity; right columns show averages over all linear layers.

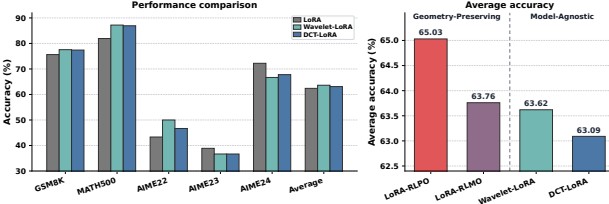

*Figure 8.* Left: Performance comparison of orthonormal LoRA variants with standard LoRA. Right: Average accuracy comparison between geometry-preserving and model-agnostic methods.

intermediate KL growth. LoRA, LoRA-RLMO, and LoRA-RLPO remain in a low-KL regime throughout training. Among them, our proposed methods are particularly stable, with KL trajectories that are consistently comparable to, and even lower than that of standard LoRA. Together with their stronger final evaluation results, these observations indicate that geometry-preserving initialization supports both stable optimization and improved downstream performance in RLVR.

**Ablation: orthonormality versus SVD geometry.** The theoretical analysis implies that an orthonormal $A_0$ minimizes the approximation error to full fine-tuning. To disentangle the two ingredients of the proposed LoRA-RLPO and LoRA-RLMO: orthonormality and SVD-based geometry information, and determine whether orthonormality alone improves RLVR training or the geometry information also contributes, we introduce two model-agnostic baselines: DCT-LoRA and Wavelet-LoRA. Both of them sat-

| | LoRA | MiLoRA | LoRA-RLMO (Ours) | PiSSA | LoRA-RLPO (Ours) |
|---|---|---|---|---|---|
| $B_0$ | $\mathbf{0}$ | $U_{-r}\Sigma_{-r}^{1/2}$ | $\mathbf{0}$ | $U_r\Sigma_r^{1/2}$ | $\mathbf{0}$ |
| $A_0$ | $\mathcal{N}(0, \frac{1}{n})$ | $\Sigma_{-r}^{1/2}V_{-r}^{\top}$ | $V_{-r}^{\top}$ | $\Sigma_r^{1/2}V_r^{\top}$ | $V_r^{\top}$ |
| GSM8K$^{@1}$ | $75.64_{\pm1.25}$ | $75.41_{\pm1.35}$ | $\mathbf{76.42_{\pm0.76}}$ | $9.65_{\pm8.20}$ | $75.59_{\pm1.14}$ |
| MATH500$^{@4}$ | $81.93_{\pm7.56}$ | $76.47_{\pm6.79}$ | $86.80_{\pm2.31}$ | $13.20_{\pm9.85}$ | $\mathbf{87.33_{\pm1.81}}$ |
| AIME22$^{@32}$ | $43.33_{\pm6.67}$ | $28.89_{\pm1.92}$ | $42.22_{\pm1.92}$ | $0.00_{\pm0.00}$ | $\mathbf{46.67_{\pm3.33}}$ |
| AIME23$^{@32}$ | $38.89_{\pm5.09}$ | $30.00_{\pm3.33}$ | $41.11_{\pm5.09}$ | $0.00_{\pm0.00}$ | $\mathbf{42.22_{\pm6.94}}$ |
| AIME24$^{@32}$ | $72.22_{\pm3.85}$ | $47.78_{\pm6.94}$ | $72.22_{\pm1.92}$ | $0.00_{\pm0.00}$ | $\mathbf{73.33_{\pm0.00}}$ |
| Avg | $62.40_{\pm2.96}$ | $51.71_{\pm0.98}$ | $63.76_{\pm1.64}$ | $4.57_{\pm3.26}$ | $\mathbf{65.03_{\pm0.55}}$ |

*Table 1.* Comparisons of SVD-based LoRA initialization methods with cosine learning rate decay. Results are reported as mean$_{\pm\text{std}}$.

isfy the orthonormality condition $A_0A_0^{\top} = I_r$ but use no information from the pretrained weight matrix $W_0$. DCT-LoRA initializes $B_0 = 0$ and $A_0 = D_r$, where $D \in \mathbb{R}^{n\times n}$ is the orthonormal Discrete Cosine Transform (DCT) matrix (Ahmed et al., 2006). Its entries are

$$D_{ij} = \alpha_i \cos\left(\frac{\pi(2j+1)i}{2n}\right), \alpha_i = \begin{cases} \sqrt{1/n}, & i = 0, \\ \sqrt{2/n}, & i > 0, \end{cases}$$

for $i, j = 0, \ldots, n - 1$. Here, $D_r \in \mathbb{R}^{r\times n}$ denotes the first $r$ rows of $D$. Wavelet-LoRA initializes $B_0 = 0$ and constructs $A_0$ as a row-orthonormal matrix from a Haar-wavelet-transformed random basis. Let $G \in \mathbb{R}^{r\times n}$ be Gaussian and let $\mathcal{H}$ denote the row-wise Haar wavelet transform. With $QR = \mathcal{H}(G)^{\top}$, we set $A_0 = Q^{\top}$, so that $A_0A_0^{\top} = I_r$.

We first verify that LoRA-RLPO and LoRA-RLMO indeed leverage and preserve the geometric structure of the pretrained weights by steering the optimization toward a specific singular subspace, whereas the model-agnostic variants DCT-LoRA and Wavelet-LoRA do not. To this end, we measure the subspace similarity between the learned model parameter $A$ and the singular vectors of pretrained matrix $W_0$. For each layer, we compute the similarity as $\|AV\|_F / \|A\|_F$ and report the average across all layers. As shown in Figure 7, LoRA-RLPO maintains high similarity with the principal singular vectors throughout training, consistent with its initialization from $V_r$, and LoRA-RLMO likewise maintains high similarity with the minor singular vectors. In contrast, DCT-LoRA and Wavelet-LoRA exhibit uniformly low similarity across all singular subspaces, confirming that these model-agnostic bases do not exploit the pretrained weight geometry.

Furthermore, Figure 8 (left) shows that both DCT-LoRA and Wavelet-LoRA outperform standard LoRA on most benchmarks, confirming that orthonormality alone improves RLVR training. Figure 8 (right) shows that LoRA-RLMO achieves performance comparable to the model-agnostic methods, while LoRA-RLPO significantly outperforms all others, indicating that principal-subspace alignment provides substantial benefit beyond orthonormality alone. Thus,

both orthonormality and SVD geometry contribute to the strong performance of our methods, with the latter yielding a particularly large gain for LoRA-RLPO.

These results also suggest that learning in the principal subspace is not inherently harmful in RLVR, but it is considerably more fragile. Although both LoRA-RLPO and PiSSA are initialized in the principal singular subspace, only LoRA-RLPO remains stable and achieves the best overall performance (Table 1). Subspace choice alone therefore does not determine the outcome; rather, our results indicate that geometry-informed orthonormal initialization combined with cosine learning-rate decay is sufficient to make principal-subspace learning both stable and effective in RLVR.

## 6. Conclusion

In this work, we studied why geometry-informed LoRA variants that are effective in supervised fine-tuning can become unstable under RLVR. We identify two primary factors governing this instability: (1) *subspace geometry*, which fundamentally shapes the optimization trajectory and concentrates the energy of parameter updates in certain subspaces; and (2) *singular-value scaling*, a distinct destabilizing factor that amplifies gradient magnitudes and drives rapid violations of the KL-divergence constraint. We provide theoretical results showing that orthonormal initialization, paired with the standard LoRA choice of zero-initializing $B_0$, minimizes the approximation gap to full fine-tuning and helps control update magnitudes. Building on these insights, we propose LoRA-RLPO and LoRA-RLMO, two geometry-preserving orthonormal initialization schemes that retain useful spectral information from the pretrained weights without singular-value scaling. Empirical results on mathematical reasoning benchmarks show that these methods stabilize RLVR training and improve downstream performance. Overall, our findings suggest that orthonormal, geometry-aware initialization offers a principled and effective foundation for low-rank adaptation in RLVR.

# Acknowledgments

Laixi Shi acknowledges funding support from MERL. Ruijia Zhang thanks Chenliang Li, Qihan Liu, Pony Ma, Qingyu Yin, and the anonymous reviewers for their insightful discussions and constructive feedback, which helped improve this paper. Ruijia Zhang also thanks Mind Lab for broader and larger-scale experimental validation of the theoretical insights in this work.

# Impact Statement

This paper advances the theoretical understanding of parameter-efficient fine-tuning for large language models in reinforcement learning settings. The proposed initialization methods aim to improve RLVR training stability and reduce the computational burden of low-rank adaptation. As with other optimization methods for LLM fine-tuning, the techniques could be used in both beneficial and harmful applications depending on the underlying model and deployment context. We do not foresee negative societal consequences specific to this work beyond those generally associated with large language model development and use.

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

## A. Related Works

**Low-rank adaptation (LoRA) and its variants.** Among parameter-efficient fine-tuning (PEFT) methods, LoRA (Hu et al., 2022) and its variants have become a popular class, parameterizing weight updates as a product of two low-rank matrices while keeping the base model frozen. Numerous variants have been proposed to improve upon standard LoRA. One line modifies the optimization process: AdaLoRA (Zhang et al., 2023) adaptively allocates rank across layers based on importance scores; LoRA+ (Hayou et al., 2024) uses different learning rates for $A$ and $B$; DoRA (Liu et al., 2024a) decomposes updates into magnitude and direction components; rsLoRA (Kalajdzievski, 2023) adjusts the scaling factor to stabilize training at higher ranks; and VeRA (Kopiczko et al., 2024) shares frozen random matrices across layers to further reduce parameters. Another line improves LoRA initialization beyond the default random scheme (Hu et al., 2022; Hayou et al., 2024). SVD-based methods have drawn particular attention: PiSSA (Meng et al., 2024) initializes adapters using principal singular components, while MiLoRA (Wang et al., 2025) uses minor components. These methods achieve faster convergence and improved performance in supervised fine-tuning, but recent evaluations show that they underperform standard LoRA and exhibit instability in RLVR (Yin et al., 2025). Our work follows this line and focuses on understanding and addressing this gap through geometry-preserving orthonormal initialization.

**Orthonormality in LoRA.** Several prior works have explored the role of orthonormality in LoRA, primarily in the context of supervised fine-tuning. Zhu et al. (2024) investigate the asymmetry between the two LoRA matrices, showing that $A$ extracts features from inputs while $B$ maps these features to outputs; they further demonstrate that fixing $A$ as a random orthonormal matrix and training only $B$ outperforms standard LoRA. OLoRA (Büyükakyüz, 2024) uses Qrthogonal-Right triangular(QR) decomposition to initialize both LoRA matrices with orthonormal bases derived from the pretrained weights, achieving faster convergence on SFT tasks. From a complementary perspective, OFT (Qiu et al., 2023) and BOFT (Liu et al., 2024b) enforce orthogonality of weight updates throughout training, rather than only at initialization. However, these studies are largely empirical and confined to the SFT regime, whose learning dynamics differ substantially from those of RLVR. Our work provides the first theoretical explanation for why orthonormal initialization improves LoRA in RLVR, showing that it enables LoRA to more closely track the trajectory of full fine-tuning.

**LoRA for Reinforcement learning with verifiable rewards.** While LoRA has been extensively studied in supervised fine-tuning (Hu et al., 2022; Liu et al., 2024a; Kalajdzievski, 2023; Hayou et al., 2024), its behavior under RL-based fine-tuning remains far less understood, despite its widespread adoption for memory-efficient PPO and GRPO training on consumer hardware (Santacroce et al., 2023; Guo et al., 2025; Shao et al., 2024). Zhu et al. (2025) provides theoretical analysis showing that RLVR updates favor off-principal directions, in contrast to SFT which targets principal components, suggesting that methods designed for SFT may not transfer directly to RLVR. Yin et al. (2025) systematically evaluate PEFT methods under RLVR and find that SVD-based initializations such as PiSSA and MiLoRA underperform standard LoRA and exhibit training instability. Despite this progress, the appropriate initialization and subspace choice for LoRA in RLVR remains unsettled. Our work addresses this gap by providing a unified theoretical framework that explains both failure modes and showing that, when paired with the standard LoRA choice of $B = 0$, geometry-preserving orthonormal initialization of $A$ offers a principled practical solution.

## B. Preliminary and Additional Theoretical Results

### B.1. Review of Conservative Updates in RLVR

The conservative-update principle has been instantiated in several ways by modern RL algorithms. Throughout this subsection, $\pi_\theta$ denotes the rollout policy used to collect samples, and $\pi_{\theta+}$ denotes the candidate updated policy.

*Trust-region constraint (TRPO).* Schulman et al. (2015) directly constrains the KL divergence between consecutive policies:

$$\max_{\theta^+} \mathcal{L}_\theta(\theta^+) \qquad \text{s.t.} \qquad \bar{D}_{\text{KL}}(\theta, \theta^+) \le \delta, \tag{9}$$

where $\mathcal{L}_\theta(\theta^+) = \mathbb{E}_t \left[ \frac{\pi_{\theta+}(a_t|s_t)}{\pi_\theta(a_t|s_t)} A_t \right]$ is the importance-weighted surrogate. Here, $t$ denotes the time step, $s_t$ is the state, $a_t$ is the action, and $A_t$ is the advantage estimate computed from samples generated by $\pi_\theta$. $\bar{D}_{\text{KL}}(\theta, \theta^+)$ is the average KL divergence between $\pi_\theta$ and $\pi_{\theta+}$ over states. The trust-region radius $\delta$ directly limits how far the policy can move.

*Clipped surrogate (PPO).* Schulman et al. (2017) approximate the trust-region constraint by clipping the probability ratio

$r_t(\theta^+) = \pi_{\theta^+}(a_t \mid s_t)/\pi_\theta(a_t \mid s_t)$:

$$\mathcal{L}^{\mathrm{CLIP}}(\theta^+) = \mathbb{E}_t\Big[\min\big(r_t(\theta^+)\,A_t,\; \mathrm{clip}\big(r_t(\theta^+),\, 1-\epsilon,\, 1+\epsilon\big)\,A_t\big)\Big]. \tag{10}$$

The clipping range $[1-\epsilon,\, 1+\epsilon]$ prevents the ratio from deviating too far from 1, limiting how much the policy can change in a single update.

*KL penalty (GRPO).* GRPO (Shao et al., 2024) enforces conservatism through a KL penalty to a fixed reference model, $\beta \cdot D_{\mathrm{KL}}(\pi_{\theta^+} \,\|\, \pi_{\mathrm{ref}})$. This term penalizes the updated policy for deviating from the reference policy, providing an additional regularizing force during RLVR.

Despite different implementations, all these mechanisms enforce the same principle: each policy update must remain small enough that the training signal estimated from the rollout policy stays reliable. Any initialization that causes updates to exceed this implicit budget risks violating the conservative-update requirement and destabilizing training.

### B.2. Smoothness of the loss function (Assumption 4.1)

We verify that Assumption 4.1 holds for both supervised fine-tuning (SFT) with cross-entropy loss and reinforcement learning with verifiable rewards (RLVR) with policy gradient loss.

**Cross-entropy loss:** Let $y = \mathrm{softmax}(z)$. For cross-entropy $\mathcal{L} = -\sum_i t_i \log y_i$ with one-hot target $t$:

$$\frac{\partial \mathcal{L}}{\partial z} = y - t, \quad \frac{\partial^2 \mathcal{L}}{\partial z^2} = \mathrm{diag}(y) - yy^\top.$$

Let $H = \mathrm{diag}(y) - yy^\top$. For any $v$ with $\|v\|_2 = 1$:

$$v^\top H v = \sum_i y_i v_i^2 - \left(\sum_i y_i v_i\right)^2 \le \sum_i y_i v_i^2 \le 1,$$

where the first inequality uses $(\sum_i y_i v_i)^2 \ge 0$, and the second inequality follows from Jensen's inequality. Thus, $\|H\|_2 \le 1$ for all $z$, which implies $L \le 1$.

**RLVR loss:** The standard policy-gradient surrogate of (6) is:

$$\mathcal{L}_{\mathrm{PG}}(\theta) = -\mathbb{E}_{x \sim \mathcal{X},\, y \sim \pi_\theta(\cdot|x)} \left[A^\perp(x, y) \log \pi_\theta(y|x)\right],$$

where $A^\perp$ is an advantage estimate. The Hessian with respect to logits satisfies:

$$\frac{\partial^2 \mathcal{L}_{\mathrm{PG}}}{\partial z^2} = -A^\perp(x, y) \left(\mathrm{diag}(y) - yy^\top\right),$$

with spectral norm bounded by $|A^\perp(x, y)|$. For binary verifiable rewards $R \in \{0, 1\}$, we have $|A^\perp(x, y)| \le 1$, giving $L = 1$.

### B.3. PiSSA and MiLoRA failure analysis

*Table 2.* Initialization norms (Frobenius) for LoRA and MiLoRA on DeepSeek-R1-Distill-Qwen-1.5B, averaged across all target linear layers at rank $r = 16$.

|  | LoRA | MiLoRA |
|---|---|---|
| $\|B_0 A_0\|_F$ | 0 | 5.11 |

**Comparison with prior hypothesis.** Yin et al. (2025) attribute MiLoRA's failure to negligible initialization magnitude, arguing that small tail singular values cause the adapter to collapse. Our results on DeepSeek-R1-Distill-Qwen-1.5B do not support this explanation. Table 2 reports initialization norms on the 1.5B model. Notably, MiLoRA has $\|B_0 A_0\|_F = 5.11$,

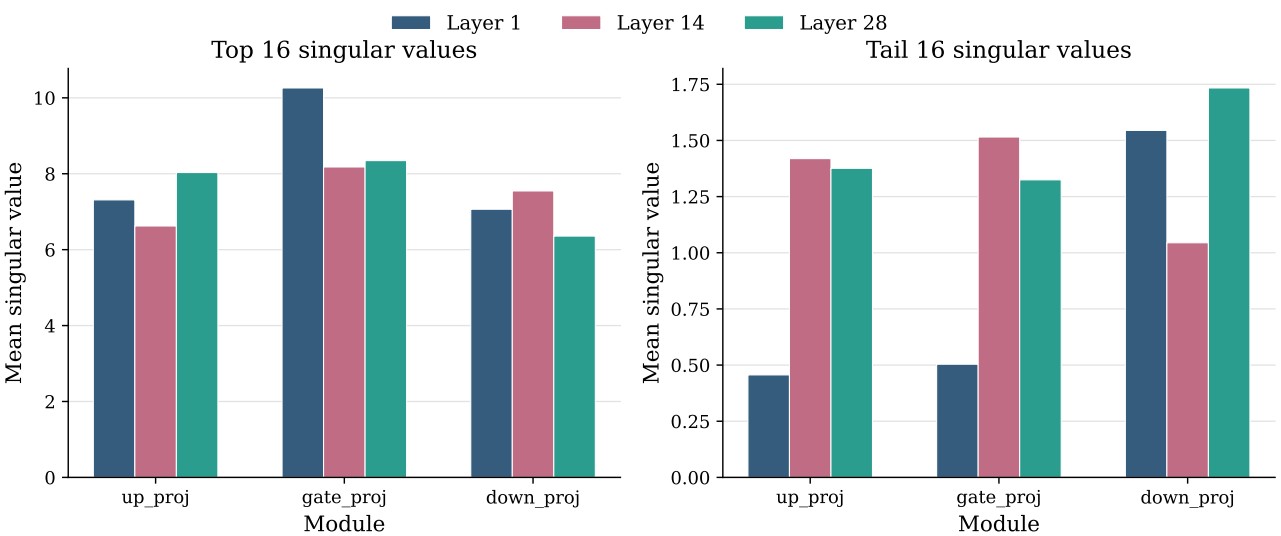

*Figure 9.* Mean singular values of MLP projection weights in DeepSeek-R1-Distill-Qwen-1.5B across representative layers 1, 14, and 28.

which is clearly non-negligible, contradicting the near-zero initialization hypothesis. In contrast, standard LoRA satisfies $\|B_0 A_0\|_F = 0$, yet it remains stable during RL training. This shows that MiLoRA's failure cannot be explained solely by initialization magnitude.

Furthermore, the singular-value structure of the 1.5B model does not support the hypothesis that MiLoRA fails because the minor subspace is numerically negligible. As shown in Figure 9, the top singular components are indeed substantially larger than the tail components across all three MLP projections, but the tail singular values remain clearly non-zero. In DeepSeek-R1-Distill-Qwen-1.5B, the mean of the tail 16 singular values is approximately 0.46–1.54 in layer 1, 1.04–1.52 in layer 14, and 1.32–1.73 in layer 28, depending on the projection module. Thus, although the minor spectrum is weaker than the principal spectrum, it still retains substantial signal across representative layers and modules. Taken together, these results suggest that MiLoRA's instability is unlikely to be caused simply by vanishing tail-spectrum magnitude.

## C. Proofs for Section 4.1 (Optimization Dynamics)

### C.1. Proof of Lemma C.1 (first-step LoRA update)

**Lemma C.1** (First-Step LoRA Update). *Let $W_0 \in \mathbb{R}^{m \times n}$ be the pretrained weight and consider the LoRA parameterization $W = W_0 + BA$ with $B_0 = 0_{m \times r}$ and $A_0 \in \mathbb{R}^{r \times n}$. Under gradient descent with step size $\eta$, the first-step weight updates satisfy:*

$$\Delta W_1^{\mathrm{LoRA}} = \Delta W_1^{\mathrm{full}} A_0^\top A_0, \tag{11}$$

*where $\Delta W_1^{\mathrm{full}} = -\eta \nabla_W \mathcal{L}(W_0)$ is the full fine-tuning update and $\Delta W_1^{\mathrm{LoRA}} = B_1 A_1$ is the LoRA update.*

*Proof.* Let $G_0 := \nabla_W \mathcal{L}(W)|_{W=W_0} \in \mathbb{R}^{m \times n}$ denote the gradient at initialization. By the chain rule for $\Delta W = BA$, the gradients with respect to the LoRA matrices are:

$$\nabla_A \mathcal{L}\big|_0 = B_0^\top G_0 = 0, \qquad \nabla_B \mathcal{L}\big|_0 = G_0 A_0^\top.$$

After one gradient step:

$$A_1 = A_0 - \eta \nabla_A \mathcal{L}\big|_0 = A_0, \qquad B_1 = B_0 - \eta \nabla_B \mathcal{L}\big|_0 = -\eta G_0 A_0^\top.$$

Thus the LoRA weight update is:

$$\Delta W_1^{\mathrm{LoRA}} = B_1 A_1 = -\eta G_0 A_0^\top A_0.$$

For full fine-tuning:

$$\Delta W_1^{\mathrm{full}} = -\eta G_0.$$

Therefore:
$$\Delta W_1^{\text{LoRA}} = \Delta W_1^{\text{full}} A_0^\top A_0,$$

which completes the proof. □

### C.2. Proof of Theorem 4.2 (LoRA approximation error)

*Proof.* Let $W_t^{\text{LoRA}}$ and $W_t^{\text{full}}$ denote the weights after $t$ gradient descent steps with step size $\eta$, initialized from the same pretrained weight
$$W_0^{\text{LoRA}} = W_0^{\text{full}} = W_0.$$

For LoRA, write
$$W_t^{\text{LoRA}} = W_0 + \Delta W_t^{\text{LoRA}}, \qquad \Delta W_t^{\text{LoRA}} = B_t A_t,$$

where $B_0 = 0$ and $A_0 \in \mathbb{R}^{r \times n}$. Define the cumulative approximation error $E_t := \left\| W_t^{\text{LoRA}} - W_t^{\text{full}} \right\|_F$. Our goal is to upper bound $E_T/T$.

Let $G_t := \nabla_W \mathcal{L}(W)\big|_{W=W_t^{\text{LoRA}}} = g_t x^\top, G_t' := \nabla_W \mathcal{L}(W)\big|_{W=W_t^{\text{full}}} = g_t' x^\top$.

By Lemma C.1, the first LoRA update satisfies
$$W_1^{\text{LoRA}} - W_0^{\text{LoRA}} = -\eta G_0 A_0^\top A_0,$$

whereas the corresponding full fine-tuning update is $W_1^{\text{full}} - W_0^{\text{full}} = -\eta G_0$.

Therefore, $E_1 = \left\| W_1^{\text{LoRA}} - W_1^{\text{full}} \right\|_F = \eta \left\| G_0(I_n - A_0^\top A_0) \right\|_F \leq \eta \|G_0\|_F \left\| I_n - A_0^\top A_0 \right\|_2$.

By Assumption 4.1, $\|G_0\|_F \leq M$, and hence $E_1 \leq M\eta \left\| I_n - A_0^\top A_0 \right\|_2$.

We next control the error recursively. For $t \geq 1$, full fine-tuning performs the update
$$W_{t+1}^{\text{full}} = W_t^{\text{full}} - \eta G_t'.$$

For LoRA, using the gradient updates
$$A_{t+1} = A_t - \eta B_t^\top G_t, \qquad B_{t+1} = B_t - \eta G_t A_t^\top,$$

and the initialization $B_0 = 0$, we have, for small $\eta$,
$$A_t = A_0 + \mathcal{O}(\eta^2), \qquad B_{t+1} - B_t = -\eta G_t A_0^\top + \mathcal{O}(\eta^3).$$

Consequently,
$$\begin{aligned}
W_{t+1}^{\text{LoRA}} - W_t^{\text{LoRA}} &= B_{t+1} A_{t+1} - B_t A_t \\
&= -\eta G_t A_0^\top A_0 + \mathcal{O}(\eta^3).
\end{aligned} \tag{12}$$

Thus,
$$\begin{aligned}
W_{t+1}^{\text{LoRA}} - W_{t+1}^{\text{full}} &= W_t^{\text{LoRA}} - W_t^{\text{full}} + \left( W_{t+1}^{\text{LoRA}} - W_t^{\text{LoRA}} \right) - \left( W_{t+1}^{\text{full}} - W_t^{\text{full}} \right) \\
&= W_t^{\text{LoRA}} - W_t^{\text{full}} - \eta G_t A_0^\top A_0 + \eta G_t' + \mathcal{O}(\eta^3) \\
&= W_t^{\text{LoRA}} - W_t^{\text{full}} - \eta G_t(A_0^\top A_0 - I_n) + \eta(G_t' - G_t) + \mathcal{O}(\eta^3).
\end{aligned} \tag{13}$$

Taking Frobenius norms and applying the triangle inequality gives
$$\begin{aligned}
E_{t+1} &\leq E_t + \eta \left\| G_t(I_n - A_0^\top A_0) \right\|_F + \eta \left\| G_t - G_t' \right\|_F + \mathcal{O}(\eta^3) \\
&\leq E_t + \eta \|G_t\|_F \left\| I_n - A_0^\top A_0 \right\|_2 + \eta \|G_t - G_t'\|_F + \mathcal{O}(\eta^3).
\end{aligned} \tag{14}$$

By Assumption 4.1, $\|G_t\|_F \leq M$. Moreover, since
$$G_t - G_t' = (g_t - g_t')x^\top,$$

we have

$$\|G_t - G'_t\|_F = \|(g_t - g'_t)x^\top\|_F = \|g_t - g'_t\|_2 \|x\|_2.$$

By Assumption 4.1, the loss is $L$-smooth with respect to the logits. Therefore,

$$\begin{aligned}
\|g_t - g'_t\|_2 &\leq L\|z_t - z'_t\|_2 \\
&= L\|(W_t^{\text{LoRA}} - W_t^{\text{full}})x\|_2 \\
&\leq L\|W_t^{\text{LoRA}} - W_t^{\text{full}}\|_F \|x\|_2 = LE_t\|x\|_2.
\end{aligned} \tag{15}$$

Substituting this into (14) yields

$$E_{t+1} \leq \left(1 + L\eta\|x\|_2^2\right) E_t + M\eta \left\|I_n - A_0^\top A_0\right\|_2 + \mathcal{O}(\eta^3). \tag{16}$$

At this point, directly unrolling (16) would produce a factor depending on $T$. Since the theorem concerns the average approximation error $E_T/T$, we equivalently write

$$E_T = \left\| \sum_{t=0}^{T-1} \left[ \left(W_{t+1}^{\text{LoRA}} - W_t^{\text{LoRA}}\right) - \left(W_{t+1}^{\text{full}} - W_t^{\text{full}}\right) \right] \right\|_F.$$

Using the triangle inequality and the same step-wise bound above, we obtain

$$\begin{aligned}
E_T &\leq \sum_{t=0}^{T-1} \left\| \left(W_{t+1}^{\text{LoRA}} - W_t^{\text{LoRA}}\right) - \left(W_{t+1}^{\text{full}} - W_t^{\text{full}}\right) \right\|_F \\
&\leq \sum_{t=0}^{T-1} \left[ M\eta \left\|I_n - A_0^\top A_0\right\|_2 + L\eta\|x\|_2^2 E_t + \mathcal{O}(\eta^2) \right].
\end{aligned} \tag{17}$$

Since $E_t \leq E_T + \mathcal{O}(\eta)$ for $t \leq T$ up to higher-order terms under small-step gradient descent, we have

$$\frac{1}{T} \sum_{t=0}^{T-1} E_t \leq \frac{1}{T} E_T + \mathcal{O}(\eta).$$

Substituting this into (17) and dividing both sides by $T$ gives

$$\frac{1}{T} E_T \leq M\eta \left\|I_n - A_0^\top A_0\right\|_2 + L\eta\|x\|_2^2 \frac{1}{T} E_T + \mathcal{O}(\eta^2).$$

Rearranging terms, provided that $L\eta\|x\|_2^2 < 1$, yields

$$\frac{1}{T} E_T \leq \frac{M\eta}{1 - L\eta\|x\|_2^2} \left\|I_n - A_0^\top A_0\right\|_2 + \mathcal{O}(\eta^2).$$

Since $E_T = \|W_T^{\text{LoRA}} - W_T^{\text{full}}\|_F$, we obtain

$$\frac{1}{T} \left\|W_T^{\text{LoRA}} - W_T^{\text{full}}\right\|_F \leq \frac{M\eta}{1 - L\eta\|x\|_2^2} \left\|I_n - A_0^\top A_0\right\|_2 + \mathcal{O}(\eta^2),$$

which completes the proof.

Let $A_0 \in \mathbb{R}^{r \times n}$ with $r < n$ and $A_0^\top A_0 \succeq 0$. Since $\text{rank}(A_0^\top A_0) \leq \text{rank}(A_0) \leq r < n$, the matrix $A_0^\top A_0$ has at least $n - r$ zero eigenvalues. Consequently, $I_n - A_0^\top A_0$ has at least $n - r$ eigenvalues equal to 1, and therefore

$$\|I_n - A_0^\top A_0\|_2 \geq 1. \tag{18}$$

Now suppose $A_0$ has orthonormal rows, i.e., $A_0 A_0^\top = I_r$. Then for any $z \in \mathbb{R}^n$ we can decompose

$$z = z_\| + z_\perp, \qquad z_\| \in \text{row}(A_0), \ z_\perp \in \text{row}(A_0)^\perp.$$

For $z_\| = A_0^\top u \in \text{row}(A_0)$,

$$A_0^\top A_0 z_\| = A_0^\top A_0 A_0^\top u = A_0^\top (A_0 A_0^\top) u = A_0^\top u = z_\|,$$

while for $z_\perp \perp \text{row}(A_0)$ we have $A_0 z_\perp = 0$ and thus $A_0^\top A_0 z_\perp = 0$. Consequently,

$$(I_n - A_0^\top A_0) z = z_\perp,$$

so

$$\|I_n - A_0^\top A_0\|_2 = \sup_{\|z\|_2=1} \|(I_n - A_0^\top A_0) z\|_2 = \sup_{\|z\|_2=1} \|z_\perp\|_2 = 1,$$

where the last equality holds because choosing $z \in \text{row}(A_0)^\perp$ gives $\|z_\perp\|_2 = 1$. Combining this with the lower bound (18) shows that the minimum possible value of $\|I_n - A_0^\top A_0\|_2$ over all $A_0 \in \mathbb{R}^{r \times n}$ is 1, and it is achieved by any row-orthonormal initialization. $\square$

### C.3. Proof of Proposition 4.3 (bounded weight updates)

*Proof.* Since $A_0$ has orthonormal rows, $A_0 A_0^\top = I_r$, which implies that $A_0^\top A_0$ is an orthogonal projection matrix onto the row space of $A_0$. In particular, $\|A_0^\top A_0\|_2 = 1$.

By Lemma C.1, the first-step LoRA update is $\Delta W_1^{\text{lora}} = -\eta G_0 A_0^\top A_0$. To bound its Frobenius norm, we use the mixed-norm submultiplicativity property: for any matrices $A \in \mathbb{R}^{m \times n}$ and $B \in \mathbb{R}^{n \times p}$,

$$\|AB\|_F \le \|A\|_F \|B\|_2.$$

Applying this inequality with $A = G_0$ and $B = A_0^\top A_0$, we obtain

$$\|\Delta W_1^{\text{lora}}\|_F = \eta \|G_0 A_0^\top A_0\|_F \le \eta \|G_0\|_F \|A_0^\top A_0\|_2 = \eta \|G_0\|_F.$$

$\square$

## D. Proof of Theorem 3.1 (PiSSA Gradient Amplification over OLoRA)

In this section, we prove Theorem 3.1, which explains why PiSSA is more aggressive than OLoRA even when both are initialized on the same principal singular subspace. We first derive the first-step update expansions for PiSSA and OLoRA, and then compare their mode-wise coefficients directly to establish a global norm amplification result.

**Lemma D.1** (PiSSA First-Step Update Expansion). *For PiSSA with principal components indexed by $\mathcal{R} = \{1, \ldots, r\}$, the first-step weight update satisfies*

$$\Delta W_1^{\text{PiSSA}} = -\eta \sum_{i,j} c_{ij}^{\text{PiSSA}} \alpha_i \beta_j \, u_i v_j^\top + \mathcal{O}(\eta^2),$$

*where $c_{ij}^{\text{PiSSA}} = \sigma_i \mathbf{1}_{i \in \mathcal{R}} + \sigma_j \mathbf{1}_{j \in \mathcal{R}}$, and $\alpha_i = u_i^\top g, \beta_j = v_j^\top x$, with $g = \frac{\partial \mathcal{L}}{\partial z}$ and $z = Wx$.*

*Proof.* We expand the gradient $G = gx^\top$ in the SVD basis as

$$G = \sum_{i,j} \alpha_i \beta_j \, u_i v_j^\top, \quad \text{where} \quad \alpha_i = u_i^\top g, \ \beta_j = v_j^\top x.$$

For PiSSA with non-zero initialization on both $A_0$ and $B_0$, the chain rule gives the first-step update

$$\Delta W_1^{\text{PiSSA}} = -\eta \left( B_0 B_0^\top G + G A_0^\top A_0 \right) + \mathcal{O}(\eta^2).$$

Substituting the PiSSA initialization $B_0 = U_r \Sigma_r^{1/2}$ and $A_0 = \Sigma_r^{1/2} V_r^\top$ yields

$$B_0 B_0^\top = U_r \Sigma_r U_r^\top, \qquad A_0^\top A_0 = V_r \Sigma_r V_r^\top.$$

Hence,

$$\Delta W_1^{\text{PiSSA}} = -\eta \left( U_r \Sigma_r U_r^\top G + G V_r \Sigma_r V_r^\top \right) + \mathcal{O}(\eta^2).$$

Substituting the SVD expansion of $G$ gives

$$\Delta W_1^{\text{PiSSA}} = -\eta \sum_{i,j} (\sigma_i \mathbf{1}_{i \in \mathcal{R}} + \sigma_j \mathbf{1}_{j \in \mathcal{R}}) \, \alpha_i \beta_j \, u_i v_j^\top + \mathcal{O}(\eta^2),$$

which completes the proof. □

**Lemma D.2** (OLoRA First-Step Update Expansion). *For OLoRA with principal components indexed by $\mathcal{R} = \{1, \ldots, r\}$, the first-step weight update satisfies*

$$\Delta W_1^{\text{OLoRA}} = -\eta \sum_{i,j} c_{ij}^{\text{OLoRA}} \alpha_i \beta_j \, u_i v_j^\top + \mathcal{O}(\eta^2),$$

*where $c_{ij}^{\text{OLoRA}} = \mathbf{1}_{i \in \mathcal{R}} + \mathbf{1}_{j \in \mathcal{R}}$, and $\alpha_i = u_i^\top g, \beta_j = v_j^\top x$, with $g = \frac{\partial \mathcal{L}}{\partial z}$ and $z = Wx$.*

*Proof.* We expand the gradient $G = gx^\top$ in the SVD basis as

$$G = \sum_{i,j} \alpha_i \beta_j \, u_i v_j^\top, \quad \text{where} \quad \alpha_i = u_i^\top g, \ \beta_j = v_j^\top x.$$

For OLoRA with non-zero initialization on both $A_0$ and $B_0$, the chain rule gives the first-step update

$$\Delta W_1^{\text{OLoRA}} = -\eta \left( B_0 B_0^\top G + G A_0^\top A_0 \right) + \mathcal{O}(\eta^2).$$

Substituting the OLoRA initialization $B_0 = U_r$ and $A_0 = V_r^\top$ yields

$$B_0 B_0^\top = U_r U_r^\top, \qquad A_0^\top A_0 = V_r V_r^\top.$$

Hence,

$$\Delta W_1^{\text{OLoRA}} = -\eta \left( U_r U_r^\top G + G V_r V_r^\top \right) + \mathcal{O}(\eta^2).$$

Substituting the SVD expansion of $G$ gives

$$\Delta W_1^{\text{OLoRA}} = -\eta \sum_{i,j} (\mathbf{1}_{i \in \mathcal{R}} + \mathbf{1}_{j \in \mathcal{R}}) \, \alpha_i \beta_j \, u_i v_j^\top + \mathcal{O}(\eta^2),$$

which completes the proof. □

**Lemma D.3** (PiSSA–OLoRA Coefficient Comparison). *For PiSSA and OLoRA initialized on the same principal components indexed by $\mathcal{R} = \{1, \ldots, r\}$, the first-step update coefficients satisfy*

$$c_{ij}^{\text{PiSSA}} \geq \sigma_r \, c_{ij}^{\text{OLoRA}} \qquad \text{for all } i, j.$$

*Proof.* By Lemma D.1 and Lemma D.2,

$$c_{ij}^{\text{PiSSA}} = \sigma_i \mathbf{1}_{i \in \mathcal{R}} + \sigma_j \mathbf{1}_{j \in \mathcal{R}}, \qquad c_{ij}^{\text{OLoRA}} = \mathbf{1}_{i \in \mathcal{R}} + \mathbf{1}_{j \in \mathcal{R}}.$$

We consider four cases.

If $i, j \in \mathcal{R}$, then

$$c_{ij}^{\text{PiSSA}} = \sigma_i + \sigma_j \geq 2\sigma_r = \sigma_r(1 + 1) = \sigma_r c_{ij}^{\text{OLoRA}}.$$

If $i \in \mathcal{R}$ and $j \notin \mathcal{R}$, then

$$c_{ij}^{\text{PiSSA}} = \sigma_i \geq \sigma_r = \sigma_r \cdot 1 = \sigma_r c_{ij}^{\text{OLoRA}}.$$

If $i \notin \mathcal{R}$ and $j \in \mathcal{R}$, then

$$c_{ij}^{\text{PiSSA}} = \sigma_j \geq \sigma_r = \sigma_r \cdot 1 = \sigma_r c_{ij}^{\text{OLoRA}}.$$

If $i, j \notin \mathcal{R}$, then

$$c_{ij}^{\text{PiSSA}} = c_{ij}^{\text{OLoRA}} = 0,$$

so the inequality also holds.

Therefore, for all $i, j$,

$$c_{ij}^{\text{PiSSA}} \geq \sigma_r \, c_{ij}^{\text{OLoRA}},$$

which completes the proof. □

*Proof of Theorem 3.1.* By Lemma D.1 and Lemma D.2, we may write

$$\Delta W_1^{\text{PiSSA}} = -\eta \sum_{i,j} c_{ij}^{\text{PiSSA}} \alpha_i \beta_j \, u_i v_j^\top + \mathcal{O}(\eta^2),$$

$$\Delta W_1^{\text{OLoRA}} = -\eta \sum_{i,j} c_{ij}^{\text{OLoRA}} \alpha_i \beta_j \, u_i v_j^\top + \mathcal{O}(\eta^2).$$

Since the matrices $\{u_i v_j^\top\}_{i,j}$ are orthonormal under the Frobenius inner product,

$$\left\| \Delta W_1^{\text{PiSSA}} \right\|_F^2 = \eta^2 \sum_{i,j} \left( c_{ij}^{\text{PiSSA}} \right)^2 \alpha_i^2 \beta_j^2 + \mathcal{O}(\eta^3),$$

$$\left\| \Delta W_1^{\text{OLoRA}} \right\|_F^2 = \eta^2 \sum_{i,j} \left( c_{ij}^{\text{OLoRA}} \right)^2 \alpha_i^2 \beta_j^2 + \mathcal{O}(\eta^3).$$

By Lemma D.3, for every $i, j$,

$$c_{ij}^{\text{PiSSA}} \geq \sigma_r \, c_{ij}^{\text{OLoRA}}.$$

Therefore,

$$\left\| \Delta W_1^{\text{PiSSA}} \right\|_F^2 \geq \eta^2 \sum_{i,j} \sigma_r^2 \left( c_{ij}^{\text{OLoRA}} \right)^2 \alpha_i^2 \beta_j^2 + \mathcal{O}(\eta^3)$$

$$= \sigma_r^2 \left\| \Delta W_1^{\text{OLoRA}} \right\|_F^2 + \mathcal{O}(\eta^3).$$

Taking square roots in the small-step regime yields

$$\left\| \Delta W_1^{\text{PiSSA}} \right\|_F \geq \sigma_r \left\| \Delta W_1^{\text{OLoRA}} \right\|_F.$$

□

*Remark* D.4. For pretrained LLMs, $\sigma_r \gg 1$ for moderate $r$ (see Figure 9). This explains why PiSSA is significantly more unstable than OLoRA in RLVR (see Figure 5): even under the same principal spectral prior, PiSSA's first-step update norm exceeds OLoRA's by a factor of at least $\Omega(\sigma_r)$, making it much more likely to violate the implicit KL budget.

## E. Experimental Details

### E.1. Training setup

**Model and dataset.** We conduct our main DAPO experiments on DeepSeek-R1-Distill-Qwen-1.5B. All methods are trained on the DAPO-Math-17k dataset, which contains 17,000 mathematical reasoning problems with verifiable answers. We apply LoRA adapters to all linear layers of the policy model, including the attention projections and MLP projections. Unless otherwise specified, all methods use the same base model, training data, reward function, and optimization setup.

**Data preprocessing.** Training prompts use the `prompt` field from the DAPO-Math-17k parquet file. We use left truncation and set the maximum prompt length to 512 tokens. During rollout generation, the maximum response length is set to 16,384 tokens, matching the long-reasoning setting used by DAPO. We sample 8 candidate responses per prompt using temperature $\tau = 1.0$, top-$p = 1.0$, and top-$k = -1$.

**Implementation.** All experiments are implemented in the verl framework. We use vLLM for rollout generation and Flash Attention for efficient attention computation. Training is run on 8 NVIDIA A100-SXM4-80GB GPUs with FSDP. The rollout engine uses tensor parallelism of size 2 during training. We enable gradient checkpointing, remove-padding optimization, chunked prefill, dynamic batch sizing, actor parameter offload, actor optimizer offload, and reference parameter offload.

**Hyperparameters.** We use two training configurations in our DAPO 1.5B experiments. The main configuration uses a constant learning-rate schedule and serves as the primary setting for comparing LoRA, PiSSA, and MiLoRA. We additionally run cosine-decay variants to study the effect of the learning-rate schedule while keeping the remaining hyperparameters matched to the corresponding constant-LR runs. Unless otherwise stated, all methods use GRPO as the advantage estimator, LoRA-style adapters applied to all linear layers, 8 responses per prompt, and the same DAPO-style objective without an explicit KL reward penalty or actor KL loss.

In the constant-LR setting, we train for 500 optimization steps using AdamW with no warmup, a constant learning-rate schedule, weight decay $0.1$, and gradient clipping at $1.0$. For the standard LoRA baseline and rank-16 variants, we use learning rate $1 \times 10^{-5}$, rank $r = 16$, and scaling parameter $\alpha = 32$, corresponding to an effective scaling factor of $\alpha/r = 2$. For the original PiSSA and MiLoRA setting, we follow the commonly used configuration with learning rate $1 \times 10^{-5}$, rank $r = 16$, and $\alpha = 32$. LoRA dropout is set to $0.0$ for all methods. The clipping range is asymmetric, with lower clip ratio $0.2$ and upper clip ratio $0.28$. The prompt batch size is 128, with a PPO mini-batch size of 32.

In the cosine-decay setting, we keep the optimizer, warmup, weight decay, batch size, rollout configuration, and adapter configuration matched to the corresponding constant-LR run, but replace the constant schedule with cosine decay. In our 1.5B cosine runs, the initial learning rate is $1 \times 10^{-5}$, warmup is 0, and weight decay is $0.1$. Table 3 summarizes the full configuration.

*Table 3.* Hyperparameters for the DAPO 1.5B experiments.

| Hyperparameter | Constant-LR setting | Cosine-decay setting |
|---|---|---|
| *Model and Software* | | |
| Base model | DeepSeek-R1-Distill-Qwen-1.5B | DeepSeek-R1-Distill-Qwen-1.5B |
| Training framework | verl 0.7.0.dev | verl 0.7.0.dev |
| Inference engine | vLLM 0.11.0 | vLLM 0.11.0 |
| Flash Attention | 2.8.1 | 2.8.1 |
| PyTorch | 2.8.0+cu126 | 2.8.0+cu126 |
| Hardware | $8 \times$ NVIDIA A100-SXM4-80GB | $8 \times$ NVIDIA A100-SXM4-80GB |
| *Optimization* | | |
| Optimizer | AdamW | AdamW |
| Learning rate | $1 \times 10^{-5}$ | $1 \times 10^{-5}$ |
| Learning rate schedule | Constant | Cosine decay |
| Warmup steps | 0 | 0 |
| Weight decay | 0.1 | 0.1 |
| Gradient clipping | 1.0 | 1.0 |
| Training steps | 500 | 500 |
| *Batch Size* | | |
| Prompt batch size | 128 | 128 |
| PPO mini-batch size | 32 | 32 |
| Responses per prompt | 8 | 8 |
| *GRPO / DAPO* | | |
| Advantage estimator | GRPO | GRPO |
| KL reward coefficient | 0.0 | 0.0 |
| Actor KL loss coefficient | 0.0 | 0.0 |
| Clip ratio lower / upper | 0.2/0.28 | 0.2/0.28 |
| Clip ratio $c$ | 10.0 | 10.0 |
| Loss aggregation | Token mean | Token mean |
| Entropy coefficient | 0 | 0 |
| Overlong buffer length | 4096 | 4096 |
| Overlong penalty factor | 1.0 | 1.0 |
| *Adapter Configuration* | | |
| Adapter type | LoRA-style | LoRA-style |
| Rank ($r$) | 16 | 16 |
| Alpha ($\alpha$) | 32 | 32 |
| Target modules | All linear layers | All linear layers |
| Dropout | 0.0 | 0.0 |
| Bias | None | None |
| *Training Rollout Generation* | | |

*Table 3.* Hyperparameters for the DAPO 1.5B experiments (continued).

| Hyperparameter | Constant-LR setting | Cosine-decay setting |
|---|---|---|
| Max prompt length | 512 | 512 |
| Max response length | 16384 | 16384 |
| Temperature | 1.0 | 1.0 |
| Top-$p$ | 1.0 | 1.0 |
| Top-$k$ | $-1$ | $-1$ |
| Rollout tensor parallel size | 2 | 2 |
| Rollout GPU memory utilization | 0.75 | 0.75 |
| Chunked prefill | Enabled | Enabled |

**Evaluation protocol.** We evaluate on GSM8K, MATH500, and AIME 2022–2025. For GSM8K, we report pass@1 with greedy decoding. For MATH500, we report pass@4 with temperature sampling. For AIME, we report pass@32. Unless otherwise stated, evaluation uses temperature $\tau = 0.6$, top-$p = 0.95$, top-$k = -1$, maximum prompt length 1024, and maximum response length 16,384. GSM8K uses greedy decoding with $\tau = 0$ and maximum response length 4096. MATH500 uses maximum response length 8192. AIME uses maximum response length 16,384.

For answer scoring, we use the DAPO math reward implementation provided in verl. GSM8K is scored with flexible numerical extraction to avoid undercounting correct answers that do not follow the strict #### format. For MATH500 and AIME, we use the default mathematical answer normalization and exact-match scoring implemented by the DAPO reward function.

### E.2. Evaluation benchmarks

We evaluate all methods on five mathematical reasoning benchmarks:

- **GSM8K** (Cobbe et al., 2021): Grade-school math word problems requiring multi-step arithmetic reasoning. We use the full test set of 1,319 examples and report pass@1 with greedy decoding.

- **MATH500** (Hendrycks et al., 2021): A 500-problem subset of the MATH benchmark covering algebra, geometry, counting, probability, number theory, and precalculus. We report pass@4.

- **AIME 2022/2023/2024**: Competition-level mathematical reasoning problems from the American Invitational Mathematics Examination. Each year contains 30 problems. We report pass@32.

*Table 4.* Evaluation settings for the DAPO 1.5B experiments.

| Benchmark | Metric | Samples | Temperature | Top-$p$ | Max response length |
|---|---|---|---|---|---|
| GSM8K | pass@1 | 1 | 0 | 1.0 | 4096 |
| MATH500 | pass@4 | 4 | 0.6 | 0.95 | 8192 |
| AIME 2022–2024 | pass@32 | 32 | 0.6 | 0.95 | 16384 |

## F. Additional Experiments

### F.1. Generalization to supervised fine-tuning (SFT)

To investigate whether the proposed geometry-preserving initializations generalize beyond RL fine-tuning, we conduct Supervised Fine-Tuning (SFT) experiments on Qwen2.5-7B-Instruct. We evaluate the methods across two distinct benchmark categories:

- **GLUE** (CoLA and MRPC): Classification tasks evaluating linguistic acceptability and paraphrase detection.

- **GSM8K**: Grade school math reasoning, evaluated using strict exact match (via the `lm-eval` harness, `gsm8k_cot` task).

As shown in Table 5, both LoRA-RLPO and LoRA-RLMO generalize effectively to the SFT paradigm, consistently outperforming standard LoRA across all three benchmarks.

*Table 5.* SFT evaluation results on Qwen2.5-7B-Instruct. Results are reported as mean $\pm$ standard deviation across 3 random seeds.

| Task | LoRA | LoRA-RLPO | LoRA-RLMO |
|---|---|---|---|
| CoLA (acc.) | $85.46 \pm 0.22$ | $86.42 \pm 0.24$ | $\mathbf{86.48 \pm 0.50}$ |
| MRPC (acc.) | $86.52 \pm 0.25$ | $\mathbf{88.48 \pm 0.25}$ | $87.91 \pm 0.51$ |
| GSM8K (strict) | $23.96 \pm 8.89$ | $29.74 \pm 0.54$ | $\mathbf{33.61 \pm 2.99}$ |

**Training dynamics.** Figure 10 illustrates the training loss curves across the three SFT tasks. Both LoRA-RLPO and LoRA-RLMO consistently converge faster and reach a lower final loss compared to standard LoRA. Notably, this performance gap is most pronounced on the GSM8K reasoning task.

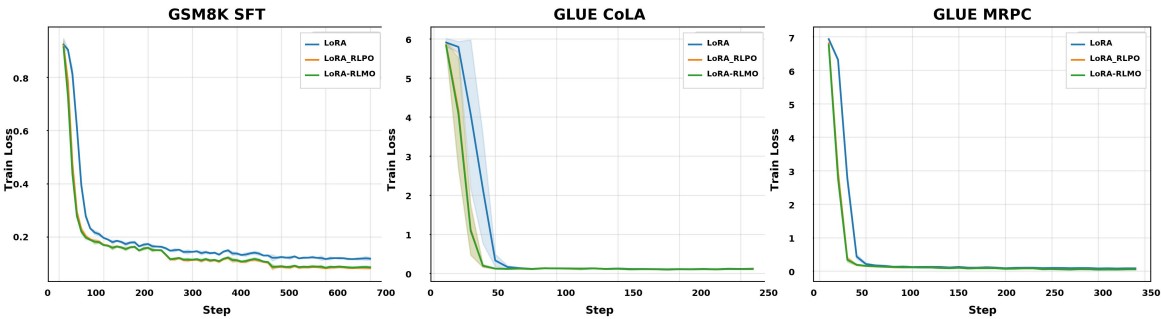

*Figure 10.* Train loss curves across three SFT tasks (3 seeds, shaded region denotes $\pm 1$ standard deviation). Both LoRA-RLPO and LoRA-RLMO demonstrate faster convergence and lower final loss than standard LoRA.

**Experimental setup.** For all SFT experiments, we use rank $r = 32$, $\alpha = 64$, and a constant learning rate of $1 \times 10^{-5}$. Models are trained for 3 epochs with a global batch size of 32 (4 per device $\times$ 8 gradient accumulation steps). All evaluations are averaged across three random seeds ($\{1, 42, 123\}$).

### F.2. SVD preprocessing cost

Table 6 reports the SVD preprocessing cost for different model scales. The measurements are conducted with rank $r = 32$ using bfloat16 precision on a single NVIDIA A100 GPU. Note that LoRA-RLPO and LoRA-RLMO have identical SVD costs. This preprocessing is a one-time cost incurred before all subsequent training steps.

*Table 6.* SVD preprocessing cost evaluated with $r = 32$, bfloat16 precision, on a single A100 GPU.

| Model | Parameters | Wall-Clock Time | Peak GPU Memory |
|---|---|---|---|
| Qwen3-4B-Instruct | 4B | 1.8 min | 8.3 GiB |
| Qwen2.5-7B-Instruct | 7B | 3.5 min | 16.0 GiB |
| Qwen2.5-14B-Instruct | 14B | 12.3 min | 29.8 GiB |

### F.3. Task domain and model family diversity.

We have added experiments on Llama 3.2-3B-Instruct (different family) and Qwen2.5-1.5B-Instruct (different scale). To test domain generality, we also evaluate on code generation: Llama 3.2-3B-Instruct on MBPP-style program synthesis with test-case-based reward. LoRA-RLPO remains stable and effective under this different reward structure.

*Table 7.* Code generation on Llama 3.2-3B-Instruct (MBPP-style, test-case reward).

| LoRA | LoRA-RLPO(Ours) | LoRA-RLMO(Ours) |
|---|---|---|
| $43.44 \pm 3.10$ | $45.67 \pm 1.41$ | $\mathbf{46.11 \pm 1.26}$ |

Our theory is loss-agnostic and applies to any RLVR objective with KL constraints.

## F.4. Larger model size

We fine-tune Qwen2.5-7B-Instruct using GRPO (Shao et al., 2024) on DAPO-Math-17k (Yu et al., 2026) with rank $r = 32$ LoRA applied to all linear layers.

**Hyperparameters.** For the 7B model experiments, we train for 150 optimization steps using the AdamW optimizer with a constant learning rate schedule and no warmup. To ensure a fair comparison, we use a learning rate of $1 \times 10^{-5}$ and scaling factor $\alpha = 64$ for standard LoRA, LoRA-RLPO, and LoRA-RLMO. For PiSSA and MiLoRA, we adopt a learning rate of $1 \times 10^{-5}$ and $\alpha = 64$, following their standard tuning practices. We set the effective batch size to 32 (4 prompts per batch with 8 responses sampled per prompt) and use a KL penalty coefficient of $\beta = 0.001$ for the GRPO objective.

As shown in Table 8, our proposed geometry-preserving initializations maintain their empirical advantages at a larger scale. Consistent with our observations on the 1.5B model, the SVD-based variants PiSSA and MiLoRA struggle under the strict KL constraints of RLVR, yielding average performances (24.74 and 26.72) that are substantially inferior to standard LoRA (30.39). In contrast, both LoRA-RLMO and LoRA-RLPO maintain stable optimization dynamics and achieve consistent performance improvements. Notably, LoRA-RLPO achieves the highest average score of $\mathbf{35.96}$ across all mathematical reasoning benchmarks, with significant gains on GSM8K and AIME evaluations. These results demonstrate that our theoretically motivated initializations scale robustly to 7B-parameter models without requiring additional hyperparameter tuning.

| | LoRA | MiLoRA | LoRA-RLMO (Ours) | PiSSA | LoRA-RLPO (Ours) |
|---|---|---|---|---|---|
| $B_0$ | $\mathbf{0}$ | $U_{-r}\Sigma_{-r}^{1/2}$ | $\mathbf{0}$ | $U_r\Sigma_r^{1/2}$ | $\mathbf{0}$ |
| $A_0$ | $\mathcal{N}(0, \frac{1}{n})$ | $\Sigma_{-r}^{1/2}V_{-r}^\top$ | $V_{-r}^\top$ | $\Sigma_r^{1/2}V_r^\top$ | $V_r^\top$ |
| GSM8K[@1] | $79.13_{\pm 4.6}$ | $57.27_{\pm 12.9}$ | $74.30_{\pm 12.3}$ | $55.60_{\pm 44.0}$ | $\mathbf{85.29}_{\pm 2.4}$ |
| MATH500[@4] | $52.80_{\pm 2.3}$ | $54.13_{\pm 1.5}$ | $\mathbf{56.73}_{\pm 1.3}$ | $50.33_{\pm 1.1}$ | $55.60_{\pm 2.6}$ |
| AIME22[@16] | $3.33_{\pm 2.7}$ | $2.22_{\pm 1.9}$ | $7.78_{\pm 1.6}$ | $4.44_{\pm 1.9}$ | $\mathbf{7.78}_{\pm 1.6}$ |
| AIME23[@16] | $11.11_{\pm 4.2}$ | $11.11_{\pm 1.9}$ | $10.00_{\pm 0.0}$ | $6.67_{\pm 3.3}$ | $\mathbf{13.33}_{\pm 2.7}$ |
| AIME24[@16] | $5.56_{\pm 1.6}$ | $8.89_{\pm 1.9}$ | $13.33_{\pm 2.7}$ | $6.67_{\pm 3.3}$ | $\mathbf{17.78}_{\pm 6.8}$ |
| Avg | 30.39 | 26.72 | 32.42 | 24.74 | $\mathbf{35.96}$ |

*Table 8.* Evaluation results of Qwen2.5-7B-Instruct fine-tuned with GRPO on DAPO-Math-17k. Our proposed LoRA-RLPO and LoRA-RLMO initializations outperform standard LoRA, whereas PiSSA and MiLoRA exhibit performance degradation.

## F.5. Learning rate sensitivity

We conduct a learning rate sweep on Qwen-2.5-7B-Instruct to evaluate the robustness of different initialization methods. Figure 11 shows the average accuracy across learning rates $\{10^{-6}, 10^{-5}, 10^{-4}\}$. LoRA-RLPO and LoRA-RLMO consistently outperform standard LoRA across all learning rates. All methods achieve peak performance at $10^{-5}$, which we adopt as the learning rate for all experiments reported in this paper.

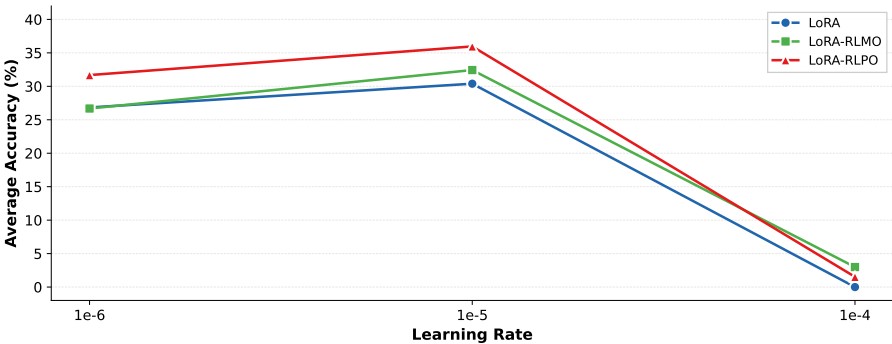

*Figure 11.* Learning rate sensitivity comparison. LoRA-RLPO and LoRA-RLMO consistently outperform standard LoRA across learning rates, with peak performance at $10^{-5}$.

