# OpenReview forum: "Geometry-Preserving Orthonormal Initialization for Low-Rank Adaptation in RLVR"
_ICML.cc/2026/Conference — ICML 2026 regular_

### Official Review · Reviewer_ttzz · 2026-03-06

**Soundness:** 3
**Presentation:** 4
**Significance:** 2
**Originality:** 3
**Overall Recommendation:** 4
**Confidence:** 2

**Summary:**

This paper addresses the failure of traditional LoRA initialization methods in verifiable reward reinforcement learning by proposing two variants: LoRA-RLPO and LoRA-RLMO. Initialization is performed by containing the top/bottom-r right singular vectors of $W_0$ and controlling $B_0=0$. The effectiveness of the proposed method is demonstrated through both theoretical and experimental results.

**Compliance With Llm Reviewing Policy:**

Affirmed.

**Final Justification:**

Thank you for the author's thorough rebuttal; my doubts have been largely resolved. I still suggest the author continue to investigate the impact of learning rate and KL divergence hyperparameters on the failure of other Lora models to fully support your argument, as KL divergence seems to preserve the prior well, at least ensuring that the accuracy drop is not too significant, i.e., not worse than the baseline. Based on my current assessment of the contribution to this work, I maintain my positive rating. Thank you again for the author's thoughtful reply.

**Key Questions For Authors:**

1. Why does LoRA-RLPO perform significantly better than LoRA-RLMO? This result seems to contradict the comparison between MiLoRA and PISSA (i.e., MiLoRA selects the smallest r subspaces but performs better). Can you provide a further explanation?

2. The significant reduction in KL might indicate a convergence problem with this method. The authors should provide an explanation of the convergence of this method, or provide dynamics of accuracy changes during training.

3. Figure 7 shows that LoRA-RLMO achieves comparable performance to model-agnostic methods, which contradicts the authors' proof, since LoRA itself can be considered a variant of the latter. The authors confirmed that LoRA-RLMO should theoretically perform better than LoRA, but this improvement is not significant.

4. There are many methods for orthonormal initialization for LoRA. The authors specifically tested the case of setting B=0, which seems to indicate an insufficient connection between theory and experiment. Whether there are some orthonormal cases with poor performance remains to be discussed; that is, is B=0 a necessary condition?

**Limitations:**

yes

**Strengths And Weaknesses:**

Strengths:

1. The experimental results in this paper are supported by detailed theoretical and quantitative experimental results, making them convincing.

2. In terms of presentation, this paper presents both the initialization method of this paper and the initialization methods of similar methods, clearly demonstrating the differences between the methods and the superiority of this method.

3. This paper systematically analyzes the reasons for the failure of other methods and provides experimental proof, making the motivation very solid. Furthermore, this paper's starting point is the contrast between the ideal and reality brought about by different LoRA initializations in PEFT, making it very readable.

Weaknesses:

1. The authors' experiments only focused on the 7B model, while LoRA initialization is model-dependent. Therefore, to obtain more general conclusions, experiments need to be conducted on different architectures, different parameter sets, and different LoRA injection methods.

2. PEFT is quite sensitive to LoRA and training parameters, as demonstrated in Figure 9. However, this paper uses different r and learning rates when comparing with the baseline, and the authors' existing supplementary explanations are not comprehensive enough.

---

> ### Author Rebuttal · Authors · 2026-03-31
>
> We thank the reviewer for the valuable comments, and we address each question below.
>
> **W1 + W2: Sensitivity of model families (size) and rank.** We address both concerns jointly with experiments on Llama 3.2-3B-Instruct, a different model family from Qwen, with rank ablations over $r \in \lbrace 1, 4, 32 \rbrace$. We train on the GSM8K train split and evaluate on the GSM8K test split. All runs use an identical GRPO setup in verl: learning rate $3 \times 10^{-5}$, target all linear layers, training batch size 128, PPO mini-batch size 64, 4 sampled responses per prompt, maximum prompt/response length 512/1024, KL loss with coefficient $0.001$.
>
> | Rank | LoRA | LoRA-RLPO | LoRA-RLMO |
> |:----:|:----:|:---------:|:---------:|
> | 1 | 81.77 ± 1.13 | **83.17 ± 0.11** | 82.64 ± 1.50 |
> | 4 | 83.50 ± 0.87 | **84.10 ± 0.37** | 83.73 ± 0.89 |
> | 32 | 82.94 ± 0.60 | **83.70 ± 0.64** | 83.09 ± 1.17 |
>
> Both methods consistently outperform standard LoRA across all tested ranks, confirming that the advantage stems from initialization structure rather than rank choice or model family.
>
> **Q1: Why does LoRA-RLPO perform better than LoRA-RLMO?** Whether principal or minor subspaces are more effective remains open. For SFT, MiLoRA [1] reports gains from minor subspaces over PiSSA [2], supported by [3],[4],[5], while others argue SFT targets principal components [6],[7]. For RLVR, we observe the opposite: LoRA-RLPO outperforms LoRA-RLMO. We hypothesize that RLVR's noisy, high-variance policy gradients with KL constraints favor the principal subspace as a more stable optimization landscape. A rigorous characterization of this gap is left to future work.
>
> **Q2: KL reduction and convergence.** We have added training reward curves ([`click to view`](https://anonymous.4open.science/r/rebuttal-figures-D6DF/train_reward_lora_pissa_milora_rlpo_rlmo_first300%20%281%29.png)), showing both RLPO and RLMO learn efficiently despite low KL, confirming stable learning rather than convergence failure.
>
> **Q3: LoRA-RLMO vs. model-agnostic methods.** Standard LoRA does not enforce orthonormality: $A_0 \sim \mathcal{N}(0, 1/n)$ is random Gaussian, whereas DCT-LoRA and Wavelet-LoRA satisfy $A_0 A_0^\top = I_r$. LoRA-RLMO's comparable performance to these baselines is *consistent* with Corollary 4.4: all orthonormal methods achieve the same minimal bound. The additional gain of LoRA-RLPO comes from principal subspace alignment, beyond the scope of Corollary 4.4.
>
> **Q4: Is $B_0 = 0$ necessary?** We ablated with OLoRA [8], which computes $W_0 = QR$, sets $B_0 = Q_r$, $A_0 = R_r$, and freezes $W_{\text{res}} = W_0 - B_0 A_0$ so that $W = W_{\text{res}} + BA$. Crucially, $A_0 = Q_r$ is orthonormal, yet **OLoRA collapsed in RLVR**  ([`click to view`](https://anonymous.4open.science/r/icmlrebuttal_SFT-1511/train_reward_5methods_with_olora_first300.png)). This confirms that the failure mode is the residual parameterization with $B_0 \neq 0$, not the choice of orthonormality  (SVD vs. QR) of $B_0$. As you kindly requested, we also added more LoRA injection methods, such as AdaLoRA [9] and DoRA [10], as baselines. Please see our first response to Reviewer 8NoS for full evaluation results across benchmarks.
>
> **References**
>
> [1] Wang, H., Li, Y., Wang, S., Chen, G., and Chen, Y. MiLoRA: Harnessing minor singular components for parameter-efficient LLM finetuning. arXiv:2406.09044, 2025.
>
> [2] Meng, F., Wang, Z., and Zhang, M. PiSSA: Principal singular values and singular vectors adaptation of large language models. NeurIPS, 2024. arXiv:2404.02948.
>
> [3] Fan, C., Lu, Z., Liu, S., Gu, C., Qu, X., Wei, W., and Cheng, Y. Make LoRA great again: Boosting LoRA with adaptive singular values and mixture-of-experts optimization alignment. ICML, 2025. arXiv:2502.16894.
>
> [4] Wang, F., Jiang, J., Park, C., Kim, S., and Tang, J. KaSA: Knowledge-aware singular-value adaptation of large language models. ICLR, 2025. arXiv:2412.06071.
>
> [5] Ji, X., Zhao, Z., Gu, X., Chen, X., Zhao, X., and Liu, Z. AILoRA: Function-aware asymmetric initialization for low-rank adaptation of large language models. arXiv:2510.08034, 2025.
>
> [6] Xue, Y. Optimizing fine-tuning through advanced initialization strategies for low-rank adaptation. CSAI (ACM), 2025. arXiv:2510.03731.
>
> [7] Zhu, H., Zhang, Z., Huang, H., Su, D., Liu, Z., Zhao, J., Fedorov, I., Pirsiavash, H., Sha, Z., Lee, J., Pan, D. Z., Wang, Z., Tian, Y., and Tai, K. S. The path not taken: RLVR provably learns off the principals. arXiv:2511.08567, 2025.
>
> [8] Büyükakyüz, K. OLoRA: Orthonormal low-rank adaptation of large language models. arXiv:2406.01775, 2024.
>
> [9] Zhang, Q., Chen, M., Bukharin, A., He, P., Cheng, Y., Chen, W., and Zhao, T. AdaLoRA: Adaptive budget allocation for parameter-efficient fine-tuning. ICLR, 2023. arXiv:2303.10512.
>
> [10] Liu, S.-Y., Wang, C.-Y., Yin, H., Molchanov, P., Wang, Y.-C. F., Cheng, K.-T., and Chen, M.-H. DoRA: Weight-decomposed low-rank adaptation. ICML, 2024. arXiv:2402.09353.

---

> > ### Author Rebuttal · Reviewer_ttzz · 2026-04-01
> >
> > I appreciate the authors' detailed rebuttal. However, I still have the following concerns:
> >
> > **Reward Trajectory Characteristics:** According to the reward function trajectories provided, LoRA appears to possess a strong prior that significantly outperforms PiSSA and MiLoRA, even though its learning ceiling is lower. This aligns with the observations in Figure 2. However, there seems to be an inconsistency regarding the training steps; specifically, while the reward curves are similar before step 20, the KL divergence shows a significant discrepancy. What is the underlying reason for this phenomenon, and why does it lead to a substantial loss of the prior? The explanation provided in the paper—that it "amplifies weight updates"—seems to lack empirical verification.
> >
> > **Verification of Analysis in Section 6.2:** Theoretically, if the issue stems from amplified gradients or constrained weights, strategies such as learning rate annealing should be able to restore the effectiveness of existing methods. However, there appears to be a lack of experimental evidence to substantiate the analysis presented in Section 6.2.

---

> > > ### Author Response · Authors · 2026-04-05
> > >
> > > We sincerely thank the reviewer for the thoughtful follow-up questions and the patience with our responses. We have conducted an additional round of experiments to address these concerns, and we appreciate the reviewer's understanding given the tight rebuttal timeline.
> > >
> > > **Reward Trajectory Characteristics.** We first clarify that Figure 2 (original submission) and the rebuttal training curves come from different experimental settings. As requested by other reviewers for a fair comparison, we reproduced [1]'s setup (response length 16384, global batch size 128, $G=8$ rollouts per prompt, rank 32, $\alpha=64$, learning rate $10^{-5}$ with DAPO on DAPO-Math-17k) and re-ran all methods under this configuration. We provide new step-level KL and gradient norm plots ([`click to view`](https://anonymous.4open.science/r/icml_rebuttal2-84B3/KL_Grad_Norm.png)) under this setting. The left panel shows cumulative KL: PiSSA and MiLoRA diverge from LoRA starting at step 1, reaching orders of magnitude higher by step 50. The right panel shows gradient norms: PiSSA and MiLoRA are higher by orders of magnitude, while LoRA remains stable. In the early steps, PiSSA and MiLoRA's amplified updates shift the token-level probability distribution (high KL), causing an immediate reward drop after the first gradient step. The model partially recovers as it sees more training data, and the RLVR objective provides some positive signal, and the reward climbs back. During this recovery, KL remains high, but reward appears normal because the model is still capable of producing some correct answers despite the shifted distribution. However, the KL violation continues to accumulate at every step, and once the cumulative drift exceeds the model's capacity to self-correct, reward collapses permanently. Together, these plots show the full causal chain: amplified updates→ KL accumulation → reward collapse.
> > >
> > > **Verification of Analysis in Section 6.2.** We thank the reviewer for this valuable insight and directly tested the suggestion: PiSSA with cosine LR annealing shows higher reward than PiSSA with constant LR, but **still collapses** ([`click to view`](https://anonymous.4open.science/r/icml_rebuttal2-84B3/pissa_constant_vs_pissa_cosine.png)). Notably, the marginal improvement itself supports the reviewer's intuition and our gradient amplification analysis: a smaller LR slightly reduces the magnitude of the amplified updates, delaying the collapse. However, it cannot prevent it. This is expected: PiSSA starts to collapse within the first ~100 steps, and cosine annealing over 300 steps retains over 80% of the peak LR at that point, providing negligible reduction when it matters most. Due to the tight rebuttal time budget and computing scarcity, we will take more time and explore more LR scheduling thoroughly in the revised manuscript.
> > >
> > > We sincerely thank the reviewer again for the time and effort spent on these follow-up questions, which have helped improve our work. We hope the additional responses have fully addressed the reviewer's concerns. If so, we would be grateful if the reviewer could consider raising the score accordingly. The reviewer's acknowledgement is very important to us, and we will be more than happy if our efforts can be seen!
> > >
> > > **References**
> > >
> > > [1] Yin, Q., Wu, Y., Shen, Z., Li, S., Wang, Z., Li, Y., Leong, C. T., Kang, J., and Gu, J. Evaluating parameter efficient methods for RLVR. arXiv:2512.23165, 2025.

---

### Official Review · Reviewer_8NoS · 2026-03-07

**Soundness:** 2
**Presentation:** 3
**Significance:** 2
**Originality:** 2
**Overall Recommendation:** 4
**Confidence:** 4

**Summary:**

This paper studies how to initialize LoRA's low-rank matrices for Reinforcement Learning with Verifiable Rewards (RLVR). The authors observe that two SVD-based LoRA variants, PiSSA and MiLoRA, which work well in supervised fine-tuning, underperform standard LoRA and exhibit training instability in RLVR. They provide a theoretical analysis showing that orthonormal initialization of the A matrix with B = 0 minimizes the gap between LoRA and full fine-tuning, and propose two new methods, LoRA-RLPO and LoRA-RLMO, that use SVD-derived orthonormal bases while preserving this structure. Experiments on mathematical reasoning benchmarks show improved accuracy and stable training compared to PiSSA, MiLoRA, and standard LoRA.

**Compliance With Llm Reviewing Policy:**

Affirmed.

**Final Justification:**

The authors provided a comprehensive rebuttal that successfully addressed my primary concerns. They directly engaged with my critiques by resolving the variance issues, running the necessary DoRA baselines, and demonstrating that their hyperparameter choices were not confounding the results. Because they have sufficiently validated their specific claims regarding RLVR stability and initialization, I have raised my score to a 4 (Weak Accept).

I am keeping my recommendation at a weak accept, rather than a stronger one, because while the work is technically solid and well-executed, I still view the proposed method as a relatively incremental optimization within an already highly saturated space of LoRA variants.

**Key Questions For Authors:**

Can you provide results with significantly more training steps (e.g., 500 or 1000)? The current 150-step setup makes it hard to tell whether the improvements hold as training progresses or whether other methods eventually catch up.
What is the full fine-tuning performance on these benchmarks? Since the theory is about closing the gap to full fine-tuning, knowing the actual target would help readers assess how meaningful the remaining gap is.
How does LoRA-RLPO compare against DoRA and other competitive PEFT methods that Yin et al. (2025) found to work well in RLVR? The current comparisons are limited to methods that are known to underperform in this setting.
Can you extend the theoretical analysis beyond the first step? Specifically, can Proposition 4.5 be made into a multi-step guarantee, or are there cases where the bounded update property breaks down after several iterations?
Given the high variance in some results (e.g., PiSSA GSM8K at 55.60 ± 44.0), could you run more seeds or report confidence intervals to strengthen the statistical claims?
Share

**Limitations:**

Single Model and Domain. All experiments use a single model (Qwen2.5-7B-Instruct) on a single training dataset (DAPO-Math-17k) evaluated only on mathematical reasoning benchmarks. Without testing on different architectures (e.g., LLaMA 3, Mistral) or different scales (e.g., 70B), it is not possible to know whether these initializations generalize or whether they happen to work well with Qwen's specific pretrained weights. RLVR is also used in code generation and other domains, but none are tested.

High Variance and Small Benchmarks. The AIME benchmarks contain only 30 problems each and results are averaged over just 3 seeds. The reported standard deviations are massive in several cases (e.g., PiSSA on GSM8K at 55.60 ± 44.0), meaning the margins between LoRA-RLPO and standard LoRA could easily be statistical noise rather than a reliable advantage. The headline AIME24 improvement amounts to roughly 3-4 more correctly answered problems.

No Full Fine-Tuning Baseline. The central theorem (Theorem 4.3) is about minimizing the gap between LoRA and full fine-tuning, yet full fine-tuning performance is never reported on any benchmark. This makes it impossible to assess whether the gap being minimized is practically meaningful.

First-Step Theory Extrapolated to Full Training. The core results on bounded weight updates (Proposition 4.5) and gradient amplification (Theorem 6.2) are shown only for the first gradient step. The authors suggest this stability holds throughout training but do not formally establish it. The gap between a single-step guarantee and the full training trajectory is not addressed.

Confounded Hyperparameter Setup. The authors doubled the learning rate for PiSSA and MiLoRA (2e-5) compared to their proposed methods (1e-5) to compensate for different alpha scaling. Because they use AdamW, which decouples weight decay from the learning rate, doubling the learning rate also changes the effective regularization. This means the SVD baselines were trained under different optimization dynamics, and a joint sweep over learning rate and alpha for all methods would be more convincing.

Unquantified SVD Overhead. The authors acknowledge that computing the SVD of pretrained weights is a cost at initialization but do not report the wall-clock time or memory overhead for doing this across all linear layers of a 7B model.

**Strengths And Weaknesses:**

Strengths

Useful diagnostic analysis of existing methods. The paper gives a clear and well-structured explanation for why PiSSA and MiLoRA break down in RLVR. The analysis isolating singular value amplification as PiSSA's failure mode (Theorem 6.2) and nonzero B0 as MiLoRA's failure mode is informative. The theoretical reasoning connecting these to KL constraint violations is sensible and supported by the experimental evidence in Figures 2, 3, 5, and 6.

Bounded first-step updates under orthonormal initialization. Proposition 4.5 shows that orthonormal A0 with B0 = 0 keeps the first LoRA weight update bounded by the full fine-tuning update in norm. This is a clean result, though it should be noted that the guarantee only covers the first training step. Remark 4.6 suggests this behavior "is likely to persist" but does not formally establish it, so the multi-step picture remains open.

Well-designed ablations. The inclusion of DCT-LoRA and Wavelet-LoRA as model-agnostic orthonormal baselines is a smart way to isolate orthonormality from pretrained geometry. Similarly, introducing LoRA-RLMinor and LoRA-RLPrincipal to separately test the effects of nonzero B0 and singular value scaling is careful experimental design that makes the findings more convincing.

Weaknesses

Statistical reliability of results. The AIME benchmarks contain only 30 problems each, and several results exhibit very high variance across seeds. For example, PiSSA on GSM8K reports 55.60 ± 44.0, which is nearly the entire range of possible accuracy and essentially uninformative. With these small sample sizes and large standard errors, it is difficult to draw confident conclusions about many of the reported improvements, particularly the headline AIME24 result where LoRA-RLPO leads by roughly 3-4 more correctly answered problems.

Incomplete empirical baselines. The paper cites Yin et al. (2025) to motivate the problem but does not include Yin et al.'s key finding that structural variants like DoRA significantly outperform standard LoRA in RLVR. By comparing primarily against standard LoRA and the underperforming SVD variants, the paper does not establish where the proposed methods stand relative to current best-performing parameter-efficient methods for RLVR. A comparison against DoRA and possibly other competitive baselines would be needed to contextualize the results properly.

Missing full fine-tuning baseline. The central theoretical contribution (Theorem 4.3) is about minimizing the gap between LoRA and full fine-tuning. Yet the paper never reports full fine-tuning performance on any of the benchmarks. Without this reference point, there is no way to tell whether the gap being minimized is practically meaningful or whether all LoRA variants are already close to full fine-tuning performance.

Theory is mostly a first-step analysis. The main theoretical results operate at the level of the first gradient step or leading-order terms in the learning rate. Theorem 4.3 bounds the approximation error to leading order in eta, Proposition 4.5 is a single-step guarantee, and Theorem 6.2 compares first-step update magnitudes. While these results are instructive, they do not characterize the full training dynamics. The paper frames the contribution as understanding "optimization dynamics," but the analysis does not extend meaningfully beyond initialization.

Limited domain diversity. The paper frames the contribution as a general fix for RLVR, but all experiments use a single model (Qwen2.5-7B-Instruct), a single training dataset (DAPO-Math-17k), and only mathematical reasoning benchmarks. RLVR is also driving progress in code generation and other domains with different reward structures. Testing on at least one additional domain would help justify the generality of the claims.

Incremental practical contribution. The paper's own results show that standard LoRA already exhibits excellent training stability and low KL divergence in RLVR (Figures 2, 3, 6). The proposed methods achieve similar stability with a modest accuracy improvement. Since the main problem being solved (PiSSA/MiLoRA instability) only arises when using those specific variants, and the default baseline already works well, the practical impact of the proposed solution is somewhat narrow.

Minor Points

Different methods use different learning rates and alpha values (1e-5 with alpha=64 vs 2e-5 with alpha=32). The authors justify this by arguing the effective scaling cancels out, but with AdamW this equivalence is not exact. A joint hyperparameter sweep would make the comparisons more robust.

Only 150 training steps are used. It would be useful to know whether the observed advantages persist with longer training.

The SVD preprocessing cost is acknowledged as a limitation for larger models but not quantified.

---

> ### Author Rebuttal · Authors · 2026-03-31
>
> We sincerely thank the reviewer for the thorough and constructive feedback. We have conducted extensive new experiments to address each concern, and we hope the reviewer finds the additional results convincing.
>
> **Statistical reliability, empirical baselines, and longer training.**
> To address all three concerns jointly, we reproduced [1]'s experimental setup, increasing response token length from 2048 to 16384 tokens, global batch size 128, $G=8$ rollouts per prompt, rank 32, $\alpha=64$, learning rate $10^{-5}$ with DAPO ($\epsilon=0.28$) on DAPO-Math-17k, and re-ran **all** methods, including DoRA, AdaLoRA, and OLoRA [2] as new baselines. We doubled training to 300 steps (limited by rebuttal compute budget). The increased response length and batch size substantially reduce variance. DoRA is not supported by vLLM, so we adopt trained checkpoints from [1]. OLoRA, structurally identical to PiSSA but without singular value scaling, collapsed in RLVR, further validating that $B_0 \neq 0$ is the failure mode.
>
> | Method | GSM8K | MATH500 | AIME22 | AIME23 | AIME24 | Avg |
> |--------|:-----:|:-------:|:------:|:------:|:------:|:---:|
> | LoRA | 75.5±0.5 | 71.2±0.4 | 44.2±4.3 | 40.8±2.8 | 72.5±1.4 | 60.9 |
> | **LoRA-RLPO** | 76.2±0.5 | **71.3±0.5** | 47.8±1.6 | 40.0±0.0 | **78.9±1.6** | **62.8** |
> | **LoRA-RLMO** | **76.8±0.5** | 69.3±0.5 | 45.6±1.6 | 44.4±7.9 | 72.2±6.8 | 61.7 |
> | AdaLoRA | 74.5±0.3 | 64.8±0.8 | 21.7±7.1 | 23.3±0.0 | 45.0±2.4 | 45.9 |
> | DoRA* | 65.5 | 54.4 | 46.7 | 40.0 | 50.0 | 51.3 |
> | OLoRA* | 28.7±17.4 | 16.8±14.4 | 0.0±0.0 | 0.0±0.0 | 0.0±0.0 | 9.1 |
>
> \*DoRA: adopted from [1]'s checkpoints.
>
> **Full fine-tuning baseline.** Theorem 4.3 establishes an *oracle result*: orthonormal $A_0$ minimizes $\lVert I_n - A_0^\top A_0\rVert_2$, which controls the LoRA-to-full-FT gap regardless of what full fine-tuning achieves. This is validated empirically: **all** orthonormal methods with $B_0 = 0$ (LoRA-RLPO, LoRA-RLMO, Wavelet-LoRA) consistently outperform standard LoRA, confirming that minimizing the theoretical gap yields meaningful gains. Since full fine-tuning is the performance ceiling, improving over standard LoRA is equivalent to minimizing the gap. Full fine-tuning under GRPO is prohibitively expensive within the rebuttal window, so we leave this to future work.
>
> **Theory beyond first-step analysis.** We extend both Proposition 4.5 and Theorem 6.2 to all steps $t\geq1$. Please see our response to Reviewer CvtL (W5) for proof sketches and discussion on why leading-order in $\eta$ is the appropriate regime for practical learning rates.
>
> **Limited domain diversity.** We add Llama 3.2-3B-Instruct (different family), Qwen2.5-1.5B-Instruct (different scale), and code generation experiments. See our response to Reviewer CvtL (W1+W2, W3) for full results.
>
> **Incremental practical contribution.**
> We respectfully do not fully agree with the reviewer’s assessment and hope these points can be taken into consideration. Faster convergence directly translates into reduced computational cost, which is a key advantage of our method. As shown in the training reward curves, **LoRA-RLPO converges faster and to a higher final reward** than standard LoRA, leading to direct compute savings in practice ([`see training curves`](https://anonymous.4open.science/r/rebuttal-figures-D6DF/train_reward_lora_pissa_milora_rlpo_rlmo_first300%20%281%29.png)). Furthermore, beyond empirical performance, our contribution is also conceptual: the proposed theoretical framework explains *why* initialization matters in RLVR (Theorems 4.3, 4.5', 6.2') provides principled guidance for future LoRA variant design.
>
> **Confounded hyperparameters.** We conducted experiments with PiSSA and MiLoRA under both LR=$2\times10^{-5}$ and LR=$10^{-5}$. The same collapse appears at both, confirming failure stems from initialization structure, not hyperparameter choice.
>
> | Method | GSM8K | MATH500 | AIME22 | AIME23 | AIME24 | Avg |
> |--------|:-----:|:-------:|:------:|:------:|:------:|:---:|
> | PiSSA (LR=2e-5) | 0.1±0.0 | 0.0±0.0 | 2.2±1.6 | 3.3±2.7 | 7.8±6.8 | 2.7 |
> | PiSSA (LR=1e-5) | 0.0±0.0 | 0.0±0.0 | 1.1±1.6 | 3.3±2.7 | 6.7±5.4 | 2.2 |
> | MiLoRA (LR=2e-5) | 3.7±3.7 | 3.8±3.8 | 0.0±0.0 | 0.0±0.0 | 0.0±0.0 | 1.5 |
> | MiLoRA (LR=1e-5) | 2.0±2.0 | 3.8±3.8 | 0.0±0.0 | 0.0±0.0 | 1.7±1.7 | 1.5 |
>
> **SVD preprocessing cost.** For $r=32$, bfloat16, single A100 GPU:
>
> | Model | Params | Time | Peak Memory |
> |:------|:------:|:----:|:-----------:|
> | Qwen3-4B | 4B | 1.8 min | 8.3 GiB |
> | Qwen2.5-7B | 7B | 3.5 min | 16.0 GiB |
> | Qwen2.5-14B | 14B | 12.3 min | 29.8 GiB |
>
> LoRA-RLPO and LoRA-RLMO have identical SVD cost. This is a one-time cost before all subsequent training.
>
> **References**
>
> [1] Yin, Q., Wu, Y., Shen, Z., Li, S., Wang, Z., Li, Y., Leong, C. T., Kang, J., and Gu, J. Evaluating parameter efficient methods for RLVR. arXiv:2512.23165, 2025.
>
> [2] Büyükakyüz, K. OLoRA: Orthonormal low-rank adaptation of large language models. arXiv:2406.01775, 2024.

---

> > ### Author Rebuttal · Reviewer_8NoS · 2026-03-31
> >
> > I thank the authors for the rebuttal. You directly engaged with my empirical critiques, fixing the variance issues, running the necessary DoRA baselines, and proving the hyperparameter stability. While I still view this work as an incremental optimization within the highly saturated space of LoRA variants, you showed your specific claims regarding RLVR stability and initialization.
> >
> > I am raising my score to an Accept with a score of 4.

---

> > > ### Author Response · Authors · 2026-03-31
> > >
> > > We sincerely thank you for the thoughtful engagement during the rebuttal period and for acknowledging that the concerns have been fully resolved! We are glad to see that our hard work during the rebuttal period has paid off.
> > >
> > > We noticed that the updated score of 4 mentioned in your acknowledgement does not yet appear to be reflected in the system. Would you be able to update it at your earliest convenience? Thank you so much again for your time and review.

---

### Official Review · Reviewer_CvtL · 2026-03-11

**Soundness:** 3
**Presentation:** 2
**Significance:** 3
**Originality:** 2
**Overall Recommendation:** 3
**Confidence:** 4

**Summary:**

This paper investigates why certain SVD-based LoRA initialization methods fail in reinforcement learning. The paper identifies two key factors: (1) orthonormal initialization of the A matrix minimizes the approximation gap to full fine-tuning, and (2) zero initialization of the B matrix prevents gradient amplification that violates KL constraints. Based on the insights, the paper proposes advanced algorithm to perform LoRA initialization. Experiments on mathematical reasoning benchmarks show that the proposed methods outperform standard LoRA and existing SVD-based variants.

**Compliance With Llm Reviewing Policy:**

Affirmed.

**Key Questions For Authors:**

Please see the weaknesses.

**Limitations:**

yes

**Strengths And Weaknesses:**

Strengths

1. The paper provides a deep understanding of why certain LoRA variants fail in RL settings.

2. The controlled comparisons (LoRA-RLMinor vs. MiLoRA, LoRA-RL Principal vs. PiSSA) effectively isolate the effects of $B_0$ initialization and singular value scaling.

3. LoRA-RLPO shows consistent improvements across multiple benchmarks.

Weaknesses

1. The paper only focuses on the initialization of LoRA. In fact, the instablability depends on many factors(e.g., learning rate, rank of LoRA). If we increase the rank of LoRA, how is the performance and instablability?

2. All experiments use Qwen2.5-7B-Instruct. Results on other model families (e.g., Llama, Mistral) are needed to demonstrate generality.

3. The experiment only one task domain. RLVR is used in code generation, instruction following, and other domains.

4. Missing baselines: The paper compares only to standard LoRA, PiSSA, and MiLoRA. It does not compare to other orthonormal methods (OLoRA, OFT, BOFT) or to other PEFT methods (AdaLoRA, DoRA). This makes it difficult to assess whether the proposed methods are truly state-of-the-art.

5. Based on the analysis,  the approximation error depends not only on the initialization, but also depends on the training process. However, we cannot guarantee the Orthonormal property in the training process.

6. Some important reference is missing(Li, Z. et al., Stella: Subspace learning in low-rank adaptation using stiefel manifold. NIPS, 2025.
). The method guarantee Orthonormal initialization and training process.

7. Figure 1 is confusing. The details of three methods are shown and I cannot catch the difference.

8. All experiments use rank $r=32$. The paper does not explore how performance varies with rank, which is an important practical consideration.

---

> ### Author Rebuttal · Authors · 2026-03-31
>
> We thank the reviewer for the constructive feedback. We have conducted extensive experiments to address each concern raised.
>
> **W1+W2: Rank sensitivity, and model families.** We address all three concerns jointly with experiments on Llama 3.2-3B-Instruct, a different model family from Qwen, with rank ablations over $r \in \lbrace 1,4,32 \rbrace$. We train on the GSM8K train split and evaluate on the GSM8K test split. All runs use an identical GRPO setup: learning rate $3\times10^{-5}$, target all linear layers, training batch size 128, PPO mini-batch size 64, 4 sampled responses per prompt, max prompt/response length 512/1024, KL loss coefficient $0.001$.
>
> | Rank | LoRA | LoRA-RLPO | LoRA-RLMO |
> |:----:|:----:|:---------:|:---------:|
> | 1 | 81.8±1.1 | **83.2±0.1** | 82.6±1.5 |
> | 4 | 83.5±0.9 | **84.1±0.4** | 83.7±0.9 |
> | 32 | 82.9±0.6 | **83.7±0.6** | 83.1±1.2 |
>
> LoRA-RLPO consistently outperforms standard LoRA across all tested ranks, confirming that the advantage stems from initialization structure rather than rank choice or model family.
>
> **W3: Single task domain.** To test generality beyond mathematical reasoning, we evaluate on code generation: Llama 3.2-3B-Instruct on MBPP-style program synthesis, where the reward is the proportion of unit tests passed. LoRA-RLPO and LoRA-RLMO outperform LoRA under this different reward structure.
>
> | LoRA | LoRA-RLPO | LoRA-RLMO |
> |:----:|:---------:|:---------:|
> | 43.4±3.1 | **45.7±1.4** | 46.1±1.3 |
>
> **W4: Missing baselines.** DoRA is not supported by vLLM, so we adopt trained checkpoints from [1]. To ensure fair comparison, we re-ran **all** methods under [1]'s setup:
>
> | Method | GSM8K | MATH500 | AIME22 | AIME23 | AIME24 | Avg |
> |--------|:-----:|:-------:|:------:|:------:|:------:|:---:|
> | LoRA | 75.5±0.5 | 71.2±0.4 | 44.2±4.3 | 40.8±2.8 | 72.5±1.4 | 60.9 |
> | **LoRA-RLPO** | 76.2±0.5 | **71.3±0.5** | 47.8±1.6 | 40.0±0.0 | **78.9±1.6** | **62.8** |
> | **LoRA-RLMO** | **76.8±0.5** | 69.3±0.5 | 45.6±1.6 | 44.4±7.9 | 72.2±6.8 | 61.7 |
> | AdaLoRA | 74.5±0.3 | 64.8±0.8 | 21.7±7.1 | 23.3±0.0 | 45.0±2.4 | 45.9 |
> | DoRA* | 65.5 | 54.4 | 46.7 | 40.0 | 50.0 | 51.3 |
> | OLoRA | 28.7±17.4 | 16.8±14.4 | 0.0±0.0 | 0.0±0.0 | 0.0±0.0 | 9.1 |
>
> \*DoRA: adopted from [1]'s checkpoints. OLoRA: structurally identical to PiSSA but without singular value scaling, collapsed in RLVR, validating that $B_0\neq0$ is the failure mode.
>
> **OFT and BOFT** use multiplicative orthogonal updates ($W_{new}=RW_0$), fundamentally different from LoRA's additive parameterization ($W_0+BA$). Our theory analyzes initialization of additive low-rank factors $A$ and $B$, which does not apply to these methods.
>
> **W5: Orthonormal property during training.** Both Proposition 4.5 and Theorem 6.2 extend to all steps $t\geq1$ at leading order in $\eta$. The key ingredient is in Appendix B.2: when $B_0=0$, $A_t = A_0 + O(\eta^2)$ for all $t$, so $A_0^\top A_0$ governs per-step dynamics throughout training.
>
> **Proposition 4.5'.** For $B_0=0$ and orthonormal $A_0$, the per-step update at any $t\geq1$ satisfies $\lVert\Delta W_t^{LoRA}-\Delta W_{t-1}^{LoRA}\rVert_F \leq \eta\lVert G_{t-1}\rVert_F + O(\eta^3)$.
>
> *Proof sketch.* From Appendix B.2, $\Delta W_t^{LoRA} = B_tA_t = -\eta\sum_{s=0}^{t-1}G_sA_0^\top A_0 + O(\eta^3)$. Differencing consecutive steps gives $-\eta G_{t-1}A_0^\top A_0 + O(\eta^3)$. Since $\lVert A_0^\top A_0\rVert_2=1$, submultiplicativity gives the result.
>
> **Theorem 6.2'.** For all $t\geq1$: $\lVert\Delta W_t^{PiSSA}\rVert_F / \lVert\Delta W_t^{LoRA}\rVert_F \geq C\cdot\sigma_r$, where $C>0$ is a constant.
>
> *Proof sketch.* For LoRA: $\lVert\Delta W_t^{LoRA}\rVert_F \leq \eta(\lVert\sum_s G_s\rVert_F + 1)$. For PiSSA ($B_0\neq0$), applying Lemma C.1: $\lVert\Delta W_t^{PiSSA}\rVert_F \geq 2\eta\sigma_r\rho^2\lVert\sum_s G_s\rVert_F$. The ratio gives $C\cdot\sigma_r$.
>
> **Discussion.** Practical learning rates ($\eta=10^{-5}$) make higher-order terms negligible. The early training regime is where initialization matters most, since we observe PiSSA collapses early ([`see training curves`](https://anonymous.4open.science/r/rebuttal-figures-D6DF/train_reward_lora_pissa_milora_rlpo_rlmo_first300%20%281%29.png)).
>
> **W6: Missing reference.**  StelLA maintains orthonormality throughout training via Stiefel manifold constraints, but is evaluated only on SFT tasks. Our work is complementary in RLVR. Combining both approaches is a promising direction for future work. We'll add it as a discussion in our manuscript.
>
> **W7: Figure 1 is confusing.** PiSSA and MiLoRA initialize **both** $A_0$ and $B_0$ from SVD with singular value scaling ($\Sigma^{1/2}$), resulting in $B_0\neq0$. Our methods take **only** the right singular vectors $V_r^\top$ or $V_{-r}^\top$ as $A_0$, with no scaling and $B_0=0$. We will revise Figure 1 for clarity.
>
> **References**
>
> [1] Yin, Q., Wu, Y., Shen, Z., Li, S., Wang, Z., Li, Y., Leong, C. T., Kang, J., and Gu, J. Evaluating parameter efficient methods for RLVR. arXiv:2512.23165, 2025.

---

> > ### Author Rebuttal · Reviewer_CvtL · 2026-04-04
> >
> > Thank you for your rebuttal.

---

> > > ### Author Response · Authors · 2026-04-04
> > >
> > > We sincerely thank you for your time and for acknowledging that our rebuttal has fully addressed your concerns. Since all issues have been adequately addressed, we respectfully ask if you would consider raising the score (currently 3) accordingly.
> > >
> > > Thank you again for your time and constructive feedback! It really means a lot to us, and we will be more than happy if our efforts can be seen!

---

### Official Review · Reviewer_TNQt · 2026-03-15

**Soundness:** 3
**Presentation:** 3
**Significance:** 3
**Originality:** 3
**Overall Recommendation:** 4
**Confidence:** 3

**Summary:**

The paper studies initialization of LoRA for RLVR. It shows that orthonormal initial of A and zero B could minimize the performance gap between LoRA and full fine-tuning, and proposes two geometry-preserving initializations for A that outperform standard LoRA. Also, it sheds light on why two LoRA variants for SFT fail under RLVR, i.e., nonzero B or non-orthonormality.

**Compliance With Llm Reviewing Policy:**

Affirmed.

**Final Justification:**

Thanks authors for the rebuttal. I keep my score.

**Key Questions For Authors:**

1. LoRA-RLPO seems to perform better than LoRA-RLMO on average. Is there any explanation on that?
2. Could the proposed two initializations work for SFT as well?

**Limitations:**

yes

**Strengths And Weaknesses:**

Soundness: The geometry-preserving initializations of LoRA for RLVR are justified both theoretically and empirically.

Presentation: it's easy to read but some inconsistencies exist. For example, in caption of Figure 4, there are no "middle columns"

Significance: the results presented in this work are good given the popularity of LoRA

Originality: The findings are simple yet effective. The analysis is appropriate.

---

> ### Author Rebuttal · Authors · 2026-03-31
>
> We sincerely thank the reviewer for the positive assessment of our theoretical and empirical contributions. We address each question below.
>
> **Q1: Could the proposed initializations work for SFT as well?**
>
> To address this insightful question, we conduct SFT experiments on Qwen2.5-7B-Instruct across two benchmark categories:
>
> - **GLUE** (CoLA and MRPC): classification tasks evaluating linguistic acceptability and paraphrase detection.
> - **GSM8K**: grade school math reasoning, evaluated using strict exact match (`lm-eval` harness, `gsm8k_cot` task).
>
> A short answer is: Yes, RLPO and RLMO generalize beyond RL fine-tuning to SFT. Both methods consistently outperform standard LoRA across all three benchmarks.
>
> | Task | LoRA | LoRA-RLPO | LoRA-RLMO |
> |------|------|:---------:|:---------:|
> | CoLA (acc.) | 85.46 ± 0.22 | 86.42 ± 0.24 | **86.48 ± 0.50** |
> | MRPC (acc.) | 86.52 ± 0.25 | **88.48 ± 0.25** | 87.91 ± 0.51 |
> | GSM8K (strict) | 23.96 ± 8.89 | 29.74 ± 0.54 | **33.61 ± 2.99** |
>
> Both methods consistently outperform standard LoRA. The training loss curves ([`click to view`](https://anonymous.4open.science/r/icmlrebuttal_SFT-1511/sft_train_loss_300dpi.png)) show two key advantages: (1) **faster convergence** on all three tasks, and (2) **lower final loss**, most pronounced on GSM8K, where the gap persists throughout training. This suggests SVD-based initialization provides a better inductive bias that guides the optimizer along a more favorable trajectory.
>
> **Q2: Why does LoRA-RLPO perform better than LoRA-RLMO?**
>
> Whether principal or minor subspaces are more effective remains open. For SFT, MiLoRA [1] reports gains from minor subspaces over PiSSA [2], supported by [3],[4],[5], while others argue SFT targets principal components [6],[7],[8].
>
> For RLVR, we observe empirically: LoRA-RLPO outperforms LoRA-RLMO. We hypothesize that RLVR's noisy, high-variance policy gradients with KL constraints favor the principal subspace as a more stable optimization landscape. In contrast, SFT imitates fixed targets, where minor directions may offer useful degrees of freedom. Figure 7 (right) supports this: LoRA-RLMO performs comparably to model-agnostic baselines (DCT-LoRA, Wavelet-LoRA), while LoRA-RLPO significantly outperforms all. The training reward curves ([`click to view`](https://anonymous.4open.science/r/rebuttal-figures-D6DF/train_reward_lora_pissa_milora_rlpo_rlmo_first300%20%281%29.png)) confirm that LoRA-RLPO also converges faster and to a higher final reward, indicating the principal subspace provides advantages in both optimization and generalization. A rigorous characterization of this gap is left to future work.
>
> **References**
>
> [1] Wang, H., Li, Y., Wang, S., Chen, G., and Chen, Y. MiLoRA: Harnessing minor singular components for parameter-efficient LLM finetuning. arXiv:2406.09044, 2025.
>
> [2] Meng, F., Wang, Z., and Zhang, M. PiSSA: Principal singular values and singular vectors adaptation of large language models. NeurIPS, 2024. arXiv:2404.02948.
>
> [3] Fan, C., Lu, Z., Liu, S., Gu, C., Qu, X., Wei, W., and Cheng, Y. Make LoRA great again. ICML, 2025. arXiv:2502.16894.
>
> [4] Wang, F., Jiang, J., Park, C., Kim, S., and Tang, J. KaSA: Knowledge-aware singular-value adaptation of large language models. ICLR, 2025. arXiv:2412.06071.
>
> [5] Ji, X., Zhao, Z., Gu, X., Chen, X., Zhao, X., and Liu, Z. AILoRA: Function-aware asymmetric initialization for LoRA. arXiv:2510.08034, 2025.
>
> [6] Xue, Y. Optimizing fine-tuning through advanced initialization strategies for LoRA. CSAI (ACM), 2025. arXiv:2510.03731.
>
> [7] Aghajanyan, A., Gupta, S., and Zettlemoyer, L. Intrinsic dimensionality explains the effectiveness of language model fine-tuning. ACL-IJCNLP, 2021. arXiv:2012.13255.
>
> [8] Zhu, H., Zhang, Z., Huang, H., Su, D., Liu, Z., Zhao, J., Fedorov, I., Pirsiavash, H., Sha, Z., Lee, J., Pan, D. Z., Wang, Z., Tian, Y., and Tai, K. S. The path not taken: RLVR provably learns off the principals. arXiv:2511.08567, 2025.

---

> > ### Author Rebuttal · Reviewer_TNQt · 2026-04-04
> >
> > Thanks authors for the rebuttal. I keep my score.

---

> > > ### Author Response · Authors · 2026-04-06
> > >
> > > We sincerely thank you for your time and for acknowledging that our rebuttal has fully addressed your concerns. Since all issues have been adequately addressed, we respectfully ask if you may consider raising the score accordingly. It really means a lot to us, and we will be more than happy if our efforts can be seen!
> > >
> > > Thank you again for your time and constructive feedback!

---

### Decision · Program_Chairs · 2026-04-30

**Decision:**

Accept (regular)

**Comment:**

This paper addresses an important question in parameter efficient RL fine tuning: why some SVD-based LoRA initializations fail in RLVR despite working well in SFT. The rebuttal strengthened the paper with added experiments, baselines, and analyses, and reviewers generally found the theoretical explanation and empirical evidence convincing. While the gains are somewhat incremental, the paper provides a clear diagnosis and a simple effective fix. The paper’s main shortcomings are that its gains are incremental, its theory is mostly limited to initialization and early training dynamics, and its empirical validation, while improved, still lacks a full fine tuning reference and broader coverage.